nature
microbiology
# OPEN

# A bacterial pan-genome makes gene essentiality strain-dependent and evolvable

Federico Rosconi [1], Emily Rudmann[1,5], Jien Li[1,5], Defne Surujon[1], Jon Anthony[1], Matthew Frank[2], Dakota S. Jones [3], Charles Rock [2], Jason W. Rosch [2], Christopher D. Johnston[3] and Tim van Opijnen [1,4] ✉

**Many bacterial species are represented by a pan-genome, whose genetic repertoire far outstrips that of any single bacterial genome. Here we investigate how a bacterial pan-genome might influence gene essentiality and whether essential genes that are initially critical for the survival of an organism can evolve to become non-essential. By using Transposon insertion sequencing (Tn-seq), whole-genome sequencing and RNA-seq on a set of 36 clinical *Streptococcus pneumoniae* strains representative of >68% of the species' pan-genome, we identify a species-wide 'essentialome' that can be subdivided into universal, core strain-specific and accessory essential genes. By employing 'forced-evolution experiments', we show that specific genetic changes allow bacteria to bypass essentiality. Moreover, by untangling several genetic mechanisms, we show that gene essentiality can be highly influenced by and/or be dependent on: (1) the composition of the accessory genome, (2) the accumulation of toxic intermediates, (3) functional redundancy, (4) efficient recycling of critical metabolites and (5) pathway rewiring. While this functional characterization underscores the evolvability potential of many essential genes, we also show that genes with differential essentiality remain important antimicrobial drug target candidates, as their inactivation almost always has a severe fitness cost in vivo.**

No single gene operates in isolation; instead, they form elaborate networks of interacting components from which phenotypes emerge. Most genes in a genome are non-essential[1]. However, their degree of dispensability may depend on the genetic background and/or the composition of the environment. In contrast, essential genes are critical for growth and survival under any circumstance. This is reflected in their high level of conservation within and often across species, limited genetic diversity, and high and stable expression levels[2–4]. Consequently, essential genes are considered rigid, largely immutable key components to an organism's survival, which makes them attractive as antimicrobial targets[5].

New insights are challenging this unyielding essential-gene concept. For instance, large-scale experiments in different yeast species have shown that genomic or transcriptional changes can suppress gene essentiality[6–9]. Additionally, many bacterial species are defined by a pan-genome. This means that different strains may contain distinct sets of genes, resulting in a species with a genetic repertoire that far exceeds that of any single genome. Moreover, a limited number of comparisons using Clustered Regularly Interspaced Short Palindromic Repeats Interference (CRISPRi) and Tn-seq[10,11] in *E. coli* and *Pseudomonas aeruginosa* experimentally demonstrate that essentiality is sometimes strain-specific[12,13]. These data indicate that essentiality is a fluid concept probably influenced by both the environment and the genetic background[1,6].

The human bacterial pathogen *Streptococcus pneumoniae*, a major cause of community-acquired pneumonia and mortality[14,15], is one species with a large pan-genome. Comparative sequence analysis indicates that a genome contains ~2,100 genes, while the entire species contains >4,000 genes[16,17]. By utilizing Tn-seq, we have shown that TIGR4, a common clinical and lab strain[18,19], contains >250 essential genes. However, a Tn-seq analysis we performed in three diverse *S. pneumoniae* strains[2,20] suggests that some essential genes are non-essential in at least one strain.

In this study, we assemble a detailed functional genomics dataset for a representative collection of 36 *S. pneumoniae* strains by performing: (1) single-molecule real-time sequencing (SMRT-seq), (2) RNA-seq and (3) Tn-seq. We show that the species contains 206 universally essential genes (always present and essential), 186 core strain-specific essential genes (always present but not always essential) and 128 accessory essential genes (essential when present). We show that these different essential-gene types have different (phenotypic) characteristics; we also use 'forced-evolution' experiments to determine how probable it is that a 'solution' exists for an essential gene to become non-essential. We show that while it is hard for a universal gene to lose its essentiality, core strain-specific and accessory essentials can switch to non-essentiality due to genetic background changes. Importantly, we show that differential essential genes remain important antimicrobial drug target candidates, as their deletion almost always has a fitness cost in vitro and in vivo.

## Results

**An *S. pneumoniae* pan-genome strain collection.** A pan-genome (PG) collection was assembled, consisting of 36 *S. pneumoniae* strains. To obtain a single contig for each strain, 33 were re-sequenced with PacBio (Bioproject PRJNA514780). This collection represents 16 capsule serotypes and 17 clusters of the Global Pneumococcal Sequencing Project (GPS)[21] (Supplementary Data 1). Six are commonly used experimental strains including D39 and TIGR4[18,22], while 30 strains are part of a larger *S. pneumoniae* surveillance collection[23]. Notably, these 30 strains consist of 15

[1]Biology Department, Boston College, Chestnut Hill, MA, USA. [2]Department of Infectious Diseases, St Jude Children's Research Hospital, Memphis, TN, USA. [3]Vaccine and Infectious Disease Division, Fred Hutchinson Cancer Research Center, Seattle, WA, USA. [4]Broad Institute of MIT and Harvard, Cambridge, MA, USA. [5]These authors contributed equally: Emily Rudmann, Jien Li. ✉e-mail: tvanopij@broadinstitute.org

genetically similar pairs (Supplementary Data 2), which should display similar phenotypes.

To determine the degree to which the PG collection reflects the *S. pneumoniae* pan-genome, a 'pan-genome study group' was created, consisting of 208 strains covering 64 different GPS clusters (Supplementary Data 3)[21,24–27]. Three different pan-genome analysis tools[28–30] were used to estimate the size of the pan-genome at >4,100 genes (all genes in the species) and a core genome of 1,349 genes (present in all strains), which is similar to previous estimates[16,17] (Fig. 1a, and Supplementary Data 4 and 5). In-depth analyses indicate that the 36-strain PG collection scatters evenly through a phylogenetic tree, and is representative of the *S. pneumoniae* pan-genome by covering >68% of its genetic content (Extended Data Fig. 1 and Fig. 1a).

**Differential gene essentiality.** Tn-seq was employed to determine gene essentiality across the PG collection. Tn library construction was successful in 21 out of 36 strains. Variable transformation efficiency across strains affects this success and depends on both known and unknown factors, including capsule serotype[31]. Tn-seq predicts gene essentiality on the basis of library saturation (percentage of transposon-occupied thymidine-adenosine sites) and the lack of insertions in a gene. Out of 21 libraries, 17 had >35% saturation, enabling high-confidence essentiality predictions ranging from 274 (PG29) to 379 (PG02) (Fig. 1b and Supplementary Data 6). From the high-confidence 'essential' category, the 'essentialome' is defined as consisting of 520 genes that are essential in at least one strain. Using gene cluster comparisons of essentiality predictions across strains, the essentialome is further subdivided into: (1) 206 universal essentials: core genome genes that are present and essential in each strain; (2) 186 core strain-dependent essentials: core genome genes that are present in all strains but only essential in some; and (3) 128 accessory essentials: accessory genome genes essential in strains where they are present (Fig. 1c and Supplementary Data 7). This categorization suggests that for a substantial number of genes, essentiality is genetic background-dependent.

**Characteristics of essentialome genes.** Essential genes have certain characteristics that distinguish them from non-essential genes. For instance, essential genes often have a higher and more stable expression than the average non-essential gene[1,2,4], tend to be more genetically conserved[1,3] and are enriched in functional categories that support central cellular processes[4,32]. Of the different essential-gene categories, only universal essentials fulfill these three characteristics (Extended Data Fig. 2 and Supplementary Data 8). Since gene essentiality for some genes depends on the strain background, we reasoned they would probably still trigger a growth defect in strains where they are not essential. Tn-seq was used to determine the fitness of these core strain-dependent essential genes in rich medium (Supplementary Data 9). Mutants with

Tn insertions in core strain-dependent essentials, on average, have a significant fitness defect compared with mutants with insertions in non-essential genes (Fig. 1d). Moreover, Tn-seq in a mouse lung infection model[33] shows that core strain-dependent essentials also have a significant defect in vivo compared with non-essentials. These data highlight that essential genes have characteristics that differentiate them from non-essential genes, and in some cases distinguish the universal, core strain-dependent and accessory groups from each other. Importantly, fitness defects caused by inactivation of strain-dependent essential genes show that genetic backgrounds responsible for essentiality bypass may not be able to entirely compensate for gene loss.

**Strain-dependent essentiality is a fluid and evolvable state.** The existence of genes with variable essentiality underscores that genome composition can make essentiality evolvable. To determine whether different levels of essential-gene evolvability exist, five universal and six core strain-dependent essentials were targeted in four different backgrounds to obtain knockout mutants. Transformation with a PCR product containing a drug marker fused to ~2,000 bp of DNA flanking a non-essential gene of interest will normally, with increasing amounts of DNA, generate an increasing number of knockout transformants[34]. However, during the transformation process, additional lesions may occur elsewhere in the genome. We thereby reasoned that by using more DNA, we could increase the chance of recovering an essential gene-knockout due to simultaneously occurring mutations that bypass the loss of the essential gene (Extended Data Fig. 3).

Transformation efficiency was generally much lower when targeting universal essential genes compared with non-essential genes (Fig. 1e). However, a surprisingly large number of colonies grew for 18 out of the 20 gene/strain combinations. Whole-genome sequencing (WGS) revealed that these colonies were, in fact, merodiploids, that is, bacteria containing genomes in which a wild-type copy of the essential gene is retained[35] (Extended Data Fig. 3b and Supplementary Data 10). The large amounts of DNA used for transformation probably accounts for the high merodiploid frequency. Importantly, besides the merodiploids, no other 'solutions' were found for the five universal essentials, confirming that their essentiality cannot be easily overcome.

Transforming strains in which the six core strain-dependent essential genes are non-essential yielded many WGS-confirmed transformants (Fig. 1e). In contrast, transformation of strains in which these genes are essential showed three types of results: (1) For genes SP_0185 and SP_0376, a low number of colonies were recovered, all of which were merodiploids, indicating essentiality is not easily bypassed; (2) For gene SP_1603/*cmk*, a high number of colonies were recovered. WGS confirmed clean knockouts without additional genetic changes, indicating that SP_1603 is, in fact, a non-essential gene; (3) For genes SP_1205, SP_1229 and SP_1569, a

**Fig. 1 | Pan-genome coverage and essential genes of *Streptococcus pneumoniae* PG collection. a**, Number of gene clusters belonging to the core and accessory genomes for the 208 strains study group determined by three different methods (PanX, BF-Clust and PPanGGOLiN). Orange represents how much of the accessory genome (>68%) is present in the PG collection. **b**, Number of genes called essential, non-essential, or uncertain across 21 strains, with mutant libraries ranked by saturation measured as the percentage of the total unique TA sites in the genome occupied by the 'mariner' transposon. **c**, Left: three representative genes showing the criteria used to classify genes as 'universal', 'strain-dependent' and 'accessory essentials'. Each dot represents the essentiality score obtained for a gene in a specific strain. A score >0.9916 categorizes a gene as essential and <0.0424 as non-essential. Right: the number of genes assigned to each class. **d**, Distribution of fitness effects of non-essential and strain-dependent essentials in vitro in rich medium (SDMM) and in vivo in a mouse infection model (lungs)[33]. Black lines inside each plot represent the median, and dotted lines the first and third quartiles. *P* values were obtained from Tukey's corrected one-way analysis of variance (ANOVA). **e**, Validation of gene essentiality. Transformation efficiency (TE) ratio is the ratio between the number of colony forming units (c.f.u.) obtained after transforming 1,000 ng of a PCR product targeting an essential gene and the c.f.u. obtained after transforming 1,000 ng of a PCR product that inserts into a neutral genomic region. TE ratios are shown for five universal and six strain-dependent essential genes in four different strain backgrounds. Experiments were performed in triplicate for each strain. Each dot in the panel represents a single strain's experimental replicate. Black dots indicate strains where the targeted gene is non-essential, and red dots where the gene is essential. Squares indicate transformation experiments with successful knockout recovery.

low number of colonies were recovered, consisting of a mix of true knockouts and merodiploids, suggesting that core strain-dependent essential genes can indeed lose their essentiality through background changes, which are explored in more detail below.

**cmk inactivation causes strain-dependent outcomes.** SP_1603/ *cmk*, involved in the recycling of cytidine monophosphate (CMP) to cytidine diphosphate (CDP) (Fig. 2a), has been indicated as an essential gene in *S. pneumoniae* for almost two decades, even

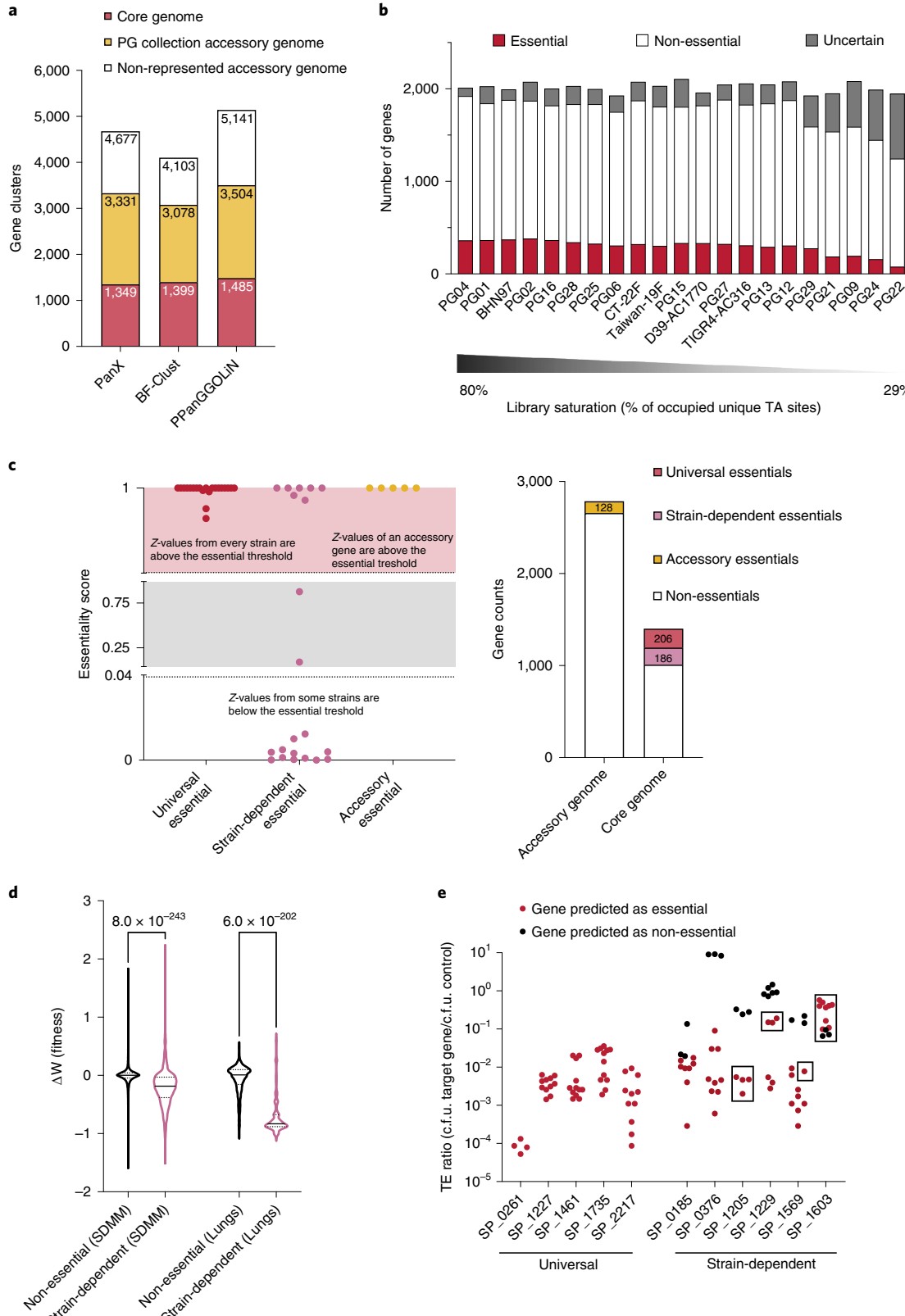

though small colonies have been observed after transformation[36]. Additionally, our Tn-seq data predict *cmk* to be essential in 12 out of 21 strains (Supplementary Data 6). However, we were able to obtain many WGS-confirmed transformants in three strain backgrounds where *cmk* was predicted as essential (Fig. 1e and Supplementary Data 10).

Growth curves of *S. pneumoniae* Δ*cmk* highlight strain-dependent differences; two out of three backgrounds display an extended lag phase, while strain PG04 has the most severe defects (Fig. 2b), probably explaining why *cmk* seems essential in Tn-seq. In *E. coli*, Δ*cmk* accumulates 30-fold more CMP and dCTP pools decrease by half[37]. However, nutritional supplementation in *S. pneumoniae* suggests that neither a dCTP shortage nor CMP accumulation reduced choline teichoic acid decoration, underlie the observed defects (Extended Data Fig. 4a).

CMP is also produced by PgsA (SP_2222), an enzyme required for phosphatidylglycerol (PtdGro) synthesis (Fig. 2c). Mutations in *pgsA* decrease PtdGro and cardiolipin (CL) and confer daptomycin resistance in four distinct Gram-positive species[38,39]. CMP accumulation, mediated by Δ*cmk*, may thus alter membrane lipid composition and daptomycin susceptibility. We examined membrane polar lipid composition by acetate-labelling and thin layer chromatography (TLC) (Fig. 2d,e and Extended Data Fig. 4b), identifying strain-specific differences. For instance, TIGR4 PtdGro levels are almost twice as high compared with other strains, which may explain why TIGR4 is so sensitive to daptomycin (Fig. 2e). Overall, Δ*cmk* decreases CL and increases galactosyl-glucosyl-diacylglycerol (Gal-Glu-DAG) levels, although changes in PG27 are modest. Previous Tn-seq on TIGR4 found that ΔSP_1075, the gene responsible for Gal-Glu-DAG synthesis (Fig. 2c), is highly sensitive to daptomycin[20]. We observed decreased daptomycin sensitivity for Δ*cmk* (Fig. 2e), which may thus be (partially) explained by increased Gal-Glu-DAG levels. There is no direct metabolic link between CMP and glycolipid synthesis, making it unclear how Δ*cmk* affects Gal-Glu-DAG levels. However, since uridine triphosphate (UTP) is the precursor to CTP, Δ*cmk* probably results in decreased UTP pools, which would need to be ramped up and thereby impact glycolipid synthesis. Importantly, this shows how something as simple as CMP accumulation can affect lipid composition, and that while *cmk* is important for growth but not essential, its effects on the processes it is connected to can vary significantly depending on the genetic background.

**Accessory essentials interact with core genome genes.** Multiple examples exist of accessory essentials interacting with core genome genes, affecting essentiality. For instance, within the genetic addiction module SP_1809/1810, the essential antitoxin (SP_1809) requires the Clp protease subunit ClpX (SP_1569) to degrade the toxin (SP_1810)[40]. *clpX* is thereby only essential in strains harbouring the toxin/antitoxin (Supplementary Data 6), as their genetic loss allows for the recovery of a *clpX* knockout in a strain where *clpX* is essential (Supplementary Data 10). Another example includes capsule-related biosynthesis genes. *S. pneumoniae*'s capsule crucially interacts with the immune system[41], which creates selective pressure that drives a high diversity in capsule serotypes. Capsule synthesis genes are organized in a large 20–30 kb operon[42], mostly consisting of accessory essentials (Fig. 3a and Supplementary Data 6). However, the sugar transferase[43] responsible for attaching the first capsule residue to the lipid carrier undecaprenyl phosphate (UP) is always non-essential. In D39, missense/nonsense mutations in the sugar transferase relieve essentiality of downstream capsule genes[44]. To determine whether this applies across the pan-genome, we knocked out the sugar transferase in four different strains and introduced transposon libraries in these query-gene backgrounds. Using Tn-seq, we find that within these sugar transferase knockout strains, the essentiality of the downstream capsule genes is indeed suppressed (Fig. 3b). This suppression of essentiality shows that blocking of the attachment of the initial UDP-sugar residue to UP prevents a lethal dead-end lipid carrier trapping[42].

As previously described, acapsular strains have an earlier onset of autolysis (Fig. 3c)[45]. Autolysis depends on the choline-binding protein LytA (SP_1739)[46], and accordingly, also relates to the amount of choline in the medium. Importantly, Tn-seq in the background of the sugar transferase knockout reveals multiple synthetic lethal interactions with teichoic acid synthesis (Fig. 3b,d, Extended Data Fig. 5 and Supplementary Data 11). These results, and the observation that unencapsulated strains present more phosphorylcholine on the cell surface[47], suggest that teichoic acid production increases in the context of the sugar transferase knockout, and this increase may explain the early onset of autolysis of *S. pneumoniae* unencapsulated strains.

Overall, these data show that essentiality of capsule biosynthesis can be bypassed by blocking its initiation. However, this interference with capsule synthesis comes at a cost, as indicated by the high number of synthetic lethal/negative interactions between the sugar transferases and other non-essential genes. Importantly, this highlights the potential of synergistic antimicrobials targeting accessory essentials and core genes.

**Strain-dependent redundancy of magnesium transporters.** *S. pneumoniae*'s core genome includes two CorA family[48] magnesium transporters (Fig. 4a; SP_0185, SP_1751). SP_0185 is essential in 8 out of 21 strains (Supplementary Data 6) and SP_1751 is non-essential. Unsuccessful attempts to construct SP_0185 knockouts in essential backgrounds suggest that bypassing its essentiality is hard. Tn-seq genetic interaction screening in PG04, a strain where the transporter is non-essential, uncovers synthetic lethality with the non-essential transporter SP_1751 (Fig. 4b and Supplementary

**Fig. 2 | Δ*cmk*/SP_1603 alters polar membrane lipid composition. a**, Schematic representation of Cmk's function in CMP recycling. Molecules in bold represent the substrate and product of the enzyme's reaction (CMP, CDP), and the ones directly involved in CMP production. *cmk* deletion affects the pathways where CDP_diacylglycerol (red and bold) is an intermediate metabolite. **b**, Growth of wild type and (complemented) SP_1603 knockouts in SDMM. **c**, Genes involved in the synthesis of the phospholipids (red tone molecules) PtdGro and CL, and the glycolipids (violet tone molecules) Glu-DAG and Gal-Glu-DAG. Molecule colours relate with bar colours in **d**. **d**, Membrane polar lipid composition of three strains and their SP_1603 knockouts. Data show existing differences in lipid composition between strains, highlighting how there is no single perfect way of building a lipid membrane. Bar graph represents the mean read counts (*n* = 2) of the different [14]C-labelled polar lipids in the membrane of the WT strains PG04, PG27 and TIGR4 (filled bars) and SP_1603 knockouts (open bars). Red bars, PtdGro counts; pink, CL; dark violet, Glu-DAG; and light violet, Gal-Glu-DAG. **e**, Growth phenotypes of the different strains grown in the presence of daptomycin. For the WT strains, increasing concentrations of daptomycin caused an extended lag phase, diminution of the growth rate and lower maximum ODs. In contrast, daptomycin only mildly affected growth of the SP_1603 knockouts. (**i**) WT growth curves in the presence of 30 μg ml⁻¹ daptomycin. (**ii**) Growth rate measured as hr⁻¹ (left) and maximum OD (right) of the different strains growing in increasing concentrations of daptomycin. (**iii**) Growth curves of the different strains at 20 (TIGR4), 30 (PG04) and 40 (PG27) μg ml⁻¹ daptomycin. (**iv**) Growth rate (left) and maximum OD (right) of the curves depicted in **iii**. *P* values were obtained from an ordinary one-way ANOVA (uncorrected Fisher's least significant difference (LSD) test). Data shown in **b** and **e** are the results from a single experiment with *n* = 3 biologically independent samples per condition. Independent repetition of each experiment in triplicate showed identical results. Error bars represent s.d.

Data 11), indicating that the essentiality of SP_0185 relies on potential functional redundancy with SP_1751.

To confirm that this redundancy lies in $Mg^{2+}$ homoeostasis, we evaluated growth of nine wild-type strains and $\Delta$SP_0185-PG04 at different $Mg^{2+}$ concentrations and in the presence of hexammine cobalt, a drug that targets CorA proteins[49] (Fig. 4c,d). Results show that: (1) $\Delta$SP_0185-PG04 acquires magnesium through another transporter sensitive to hexammine cobalt, corroborating the identified synthetic lethality with SP_1751; and (2) the

strains where SP_0185 is essential rely predominantly on one transporter to acquire $Mg^{2+}$, relinquishing SP_1751's redundancy in these strains.

To better understand the strain-dependent redundancy, we explored their genomic context, protein sequence and expression levels (Fig. 4e,f and Supplementary Table 1). First, two genomic arrangements exist for gene SP_0185 across the PG collection depending on the presence of an upstream prophage. Second, while SP_0185 is conserved, SP_1751 exhibits polymorphisms that only

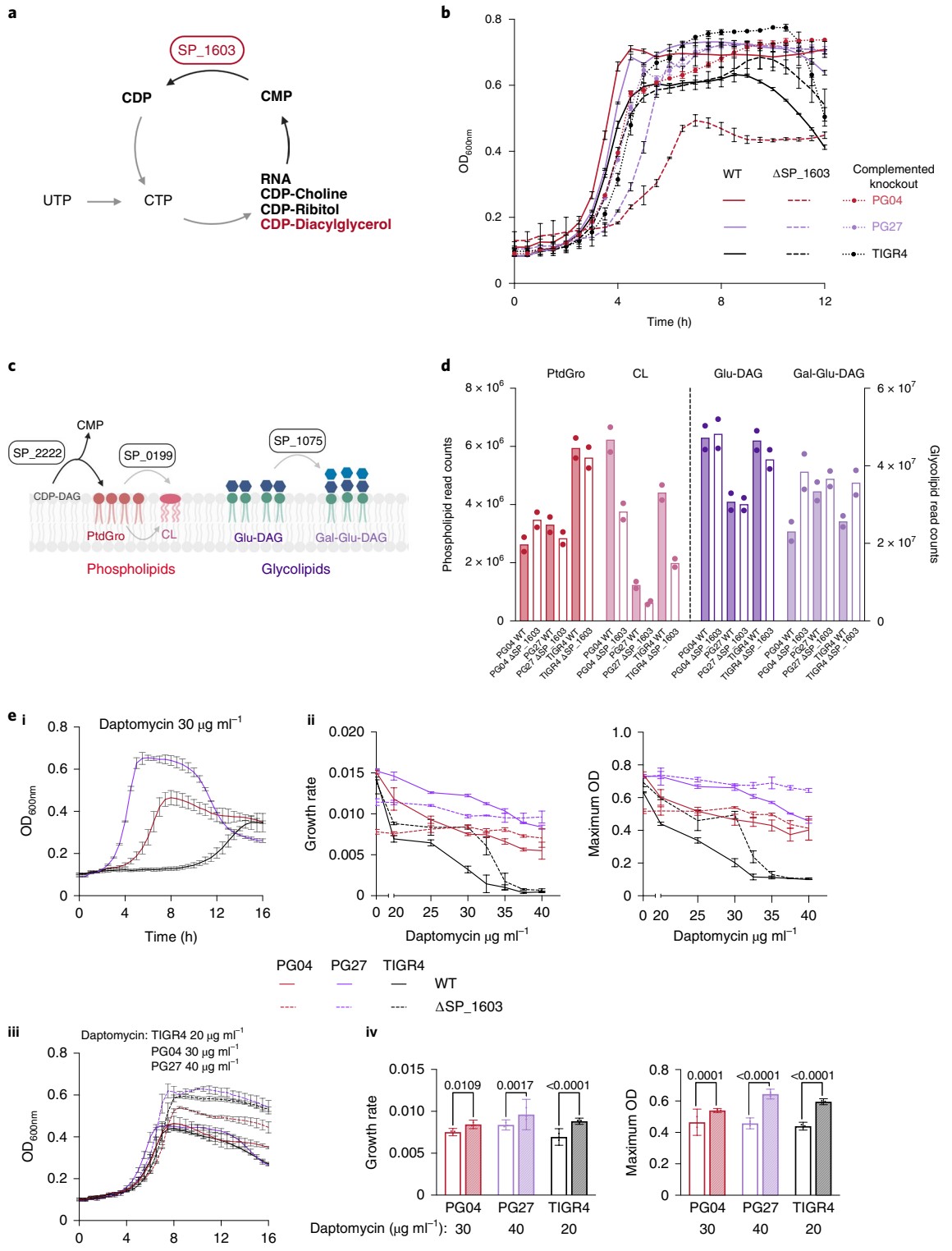

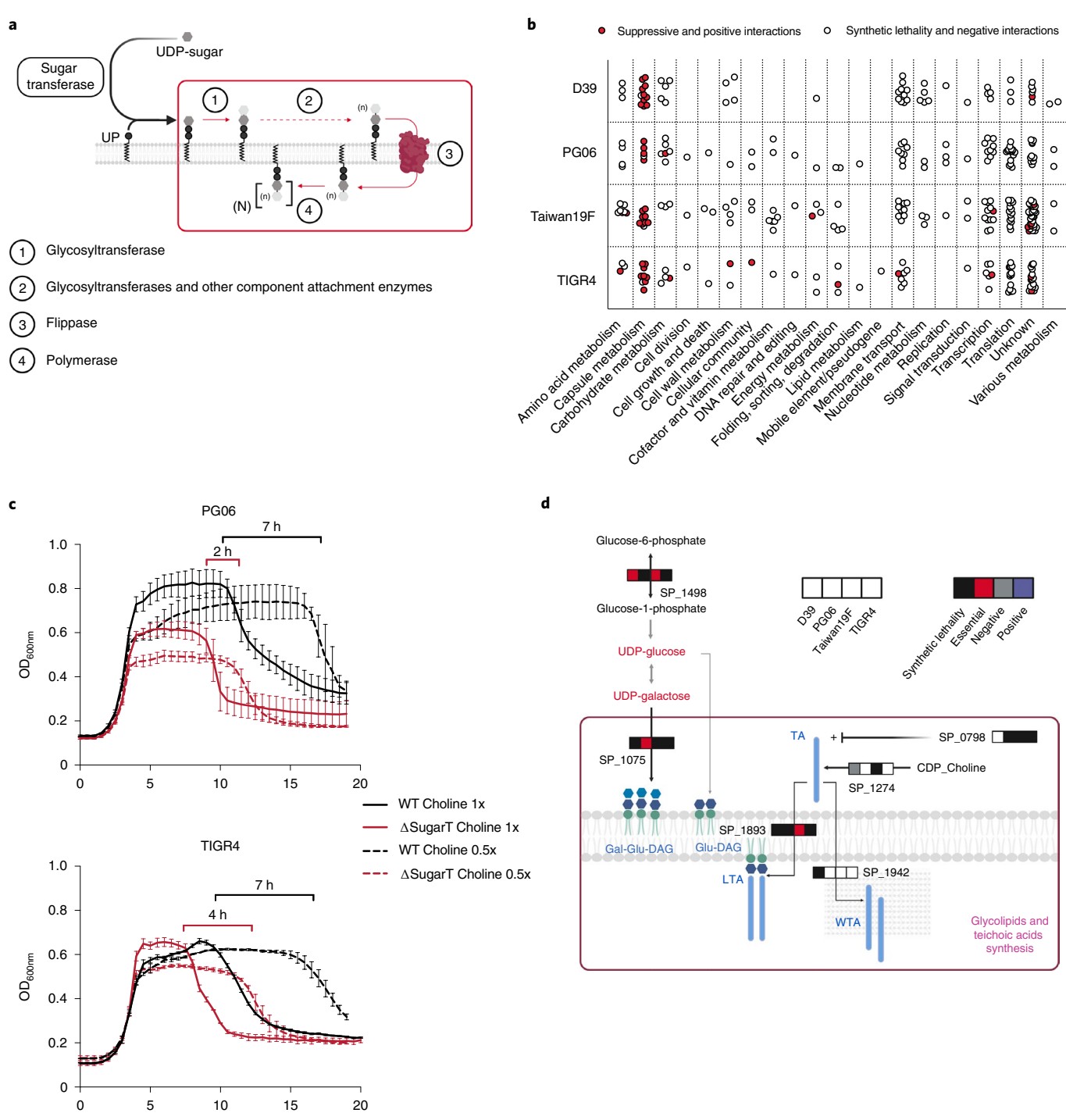

**Fig. 3 | Capsule sugar transferase characterization. a**, Schematic of capsule biosynthesis in *S. pneumoniae*. Genes inside the red box are accessory essentials, likely because the essential molecule UP gets 'trapped' if these downstream capsule genes are non-functional. **b**, Genetic interactions identified by Tn-seq using libraries constructed in the background of four different strains with a sugar transferase knockout. Each data point represents a gene; red dots, mostly in capsule metabolism, depict a suppressive interaction with the sugar transferase. These suppressor interactions are possible because locking the attachment of the initial UDP-sugar residue to UP prevents lethal dead-end lipid carrier trapping. **c**, Growth of WT strains PG06 and TIGR4 and their sugar transferase knockouts in rich medium at two choline concentrations. Autolysis in *S. pneumoniae* relies on the amount of LytA protein (SP_1937) bound to the choline moiety of wall teichoic acids[46]. Accordingly, the sugar transferase knockout-observed early autolysis relates to the amount of choline in the medium. Indicated are time differences in the onset of autolysis between the two conditions for WT (black) and knockouts (red). Shown are data from *n* = 3 biologically independent samples; independent repetition (3) showed similar results; error bars represent s.d. **d**, The sugar transferase has a synthetic lethal interaction with genes involved directly and indirectly in cell wall and teichoic acid biosynthesis. One of these interactions occurs with the response regulator CiaR (SP_0798), which upregulates teichoic acid biosynthesis genes[70]. The requirement of CiaR suggests that teichoic acid production needs to be upregulated in the context of the sugar transferase knockout. The horizontal bars of squares indicate the type of interaction identified between the sugar transferase and the indicated gene for each strain (white squares indicate no identified interaction for a specific strain). Metabolites in red (for example, UDP-Glucose) are known to accumulate in the context of acapsular *S. pneumoniae* strains[51]. LTA, lipoteichoic acid; WTA, wall teichoic acid.

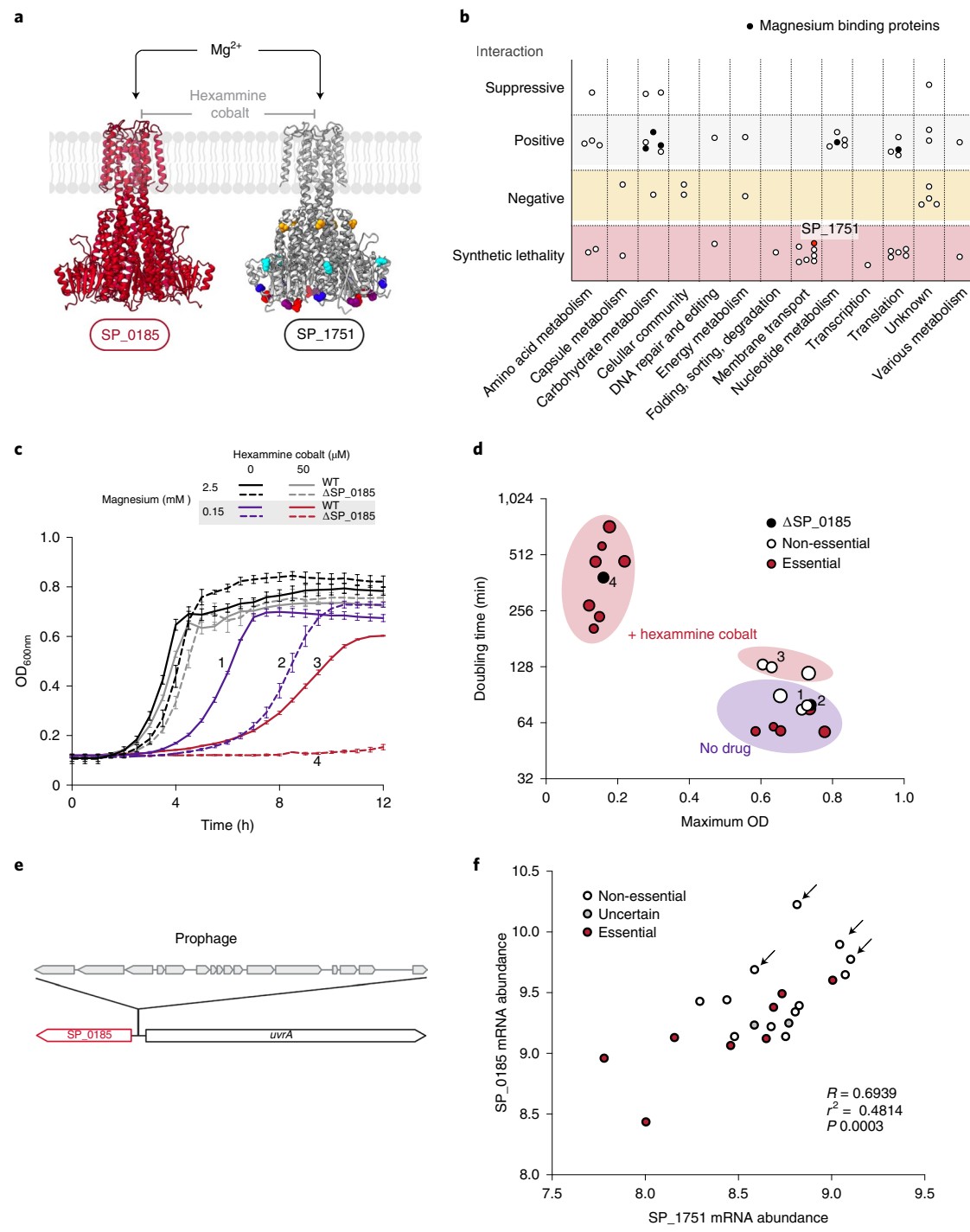

**Fig. 4 | Redundancy in magnesium transport is strain-dependent. a**, SWISS-MODEL structures of core strain-dependent essential CorA (SP_0185) (Qmean = −2.27) and the second non-essential transporter SP_1751 (Qmean = −2.82). Uniform red colour of the structure of SP_0185 indicates that the protein sequence of the transporter is fully conserved among *S. pneumoniae* strains. Non-essential transporter coded by SP_1751 conserved amino acids are represented in grey. Colours other than grey indicate residues with allelic variation across the PG collection. **b**, Genetic interactions with SP_0185 in strain PG04. Each dot represents a gene interacting with SP_0185, its type of interaction and functional category. Note that 5 out of 22 genes with a positive interaction code for magnesium-binding proteins, suggesting that preventing the use of $Mg^{2+}$ for non-essential reactions can reduce the impact of ΔSP_0185. **c**, Growth of WT PG04 and ΔSP_0185 in SDMM with different $MgCl_2$ concentrations and hexammine cobalt. Low $Mg^{2+}$ significantly affects growth of ΔSP_0185 compared with WT PG04, while addition of hexammine cobalt abolishes growth of the knockout. Data shown are from n = 3 biologically independent samples, while independent repetition (3) showed similar results. Error bars represent s.d. **d**, Doubling time versus maximum OD observed for different strains growing in rich medium with $MgCl_2$ at 0.15 mM and in the absence/presence of hexammine cobalt. Growth of WT strains in which SP_0185 is essential is similar to that of PG04-ΔSP_0185 (black dots). White and red dots represent WT strains where SP_0185 is non-essential or essential, respectively. Dots numbered 1 to 4 correspond to the numbered curves in **c**. **e**, Schematic showing the most common genomic configuration of SP_0185 and neighbouring *uvrA*, while some strains have a prophage in between. **f**, Transcript abundances of the two *corA* genes. Each dot represents a different WT strain, colours indicate SP_0185 essentiality and arrows indicate prophage presence. *R* and *P* values were derived from a least-squares linear regression.

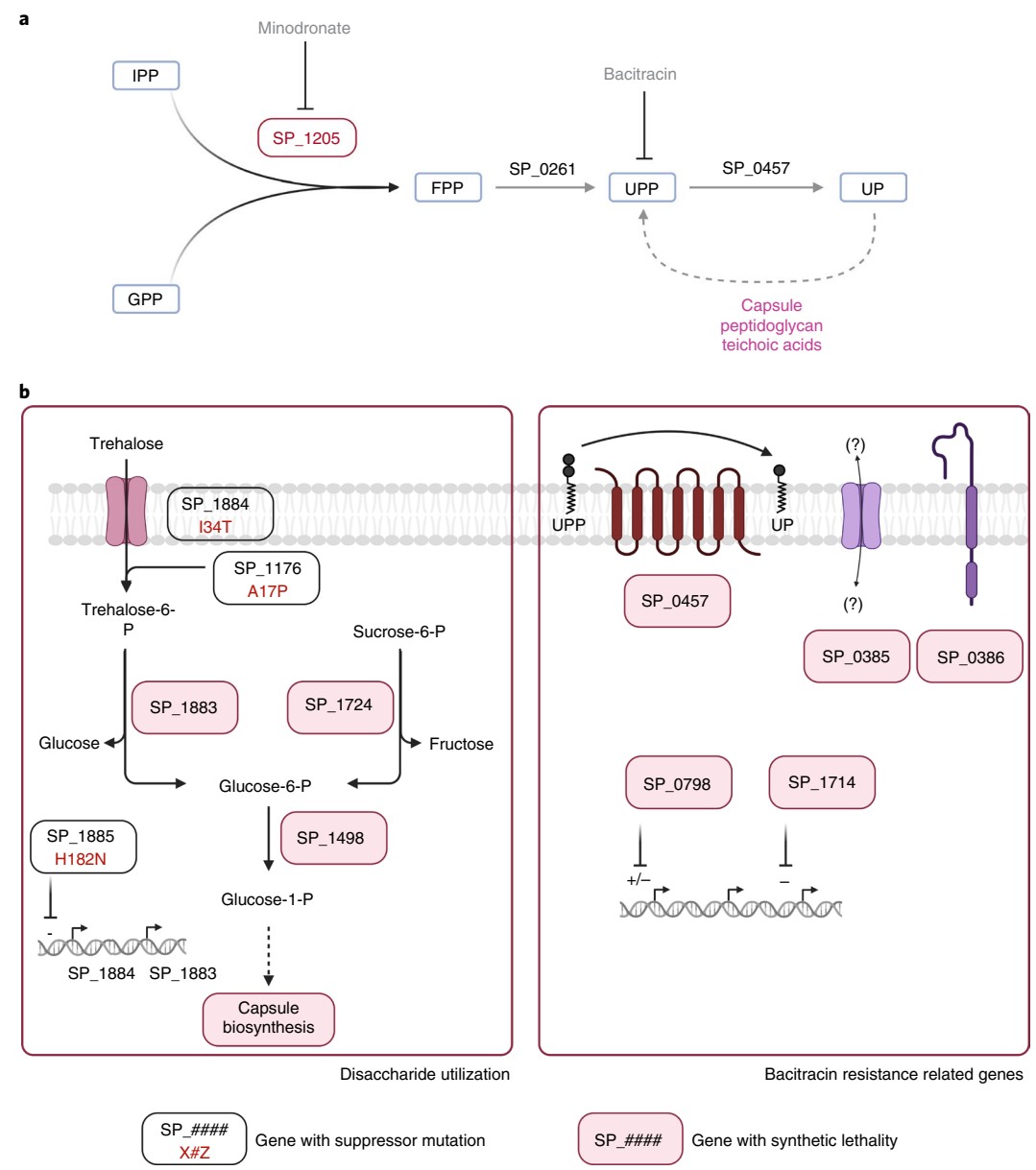

**Fig. 5 | Farnesyl-PP synthase (SP_1205) essentiality is easily bypassed. a**, Synthesis pathway of UP. The enzyme encoded by SP_1205 catalyses the condensation between isopentenyl diphosphate (IPP) and geranyl diphosphate (GPP) into farnesyl diphosphate (FPP), which is then converted to UPP by enzyme SP_0261, after which UPP is dephosphorylated by SP_0457 into its active form UP. **b**, Disaccharide metabolism and bacitracin resistance related genes are involved in bypass mechanisms of SP_1205. Suppressor mutations of SP_1205 essentiality were identified with WGS (boxes with white background) and synthetic lethality interactions (boxes with pink backgrounds) identified with Tn-seq using libraries constructed in a PG06-SP_1205 knockout. The identification of three functionally related suppressive SNPs (SP_1176 A17P, SP_1884 I34T and SP_1885 H182N) suggests that transport of the disaccharide trehalose (or a similar compound) is a central hub for bypassing SP_1205. Gene SP_1205 has a synthetic lethal interaction with two enzymes (SP_1883 and SP_1724) that hydrolyse the disaccharides trehalose and sucrose to produce glucose-6-phosphate (which feeds into glycolysis), the pentose-phosphate pathway and capsule biosynthesis (left panel). One of the genes with synthetic lethality with SP_1205 is the UPP phosphatase BacA (SP_0457). Interestingly, a TIGR4-derived SP_1205 knockout presented two compensatory mutations associated with the cell wall: one mutation is located in *murE* (SP_1530 A430S), which is involved in peptidoglycan synthesis, and the second mutation is located in a non-characterized membrane protein coding gene (SP_0454 H581N), which forms a complex with BacA[71] (right panel).

appear in strains where SP_0185 is essential. Third, strains with the prophage display some of the highest expression levels for both transporters and are the strains in which SP_0185 is non-essential. This suggests that a high expression of SP_1751 in combination with several specific polymorphisms can overcome SP_0185's essentiality. However, while the magnesium transporters are at least partially redundant in a strain-dependent manner, on the species level, SP_0185 is the main transporter.

**Farnesyl-PP-synthase essentiality bypass evolves easily.** Undecaprenyl phosphate (UP) is a crucial bacterial molecule involved in translocating peptidoglycan, teichoic acids and capsule intermediates from the cytoplasm to the outer leaf of the membrane[50]. The core strain-dependent essential gene SP_1205 forms part of the UP-biosynthesis pathway (Fig. 5a). We confirmed SP_1205's non-essentiality in PG06, while for strains PG16 and TIGR4, in which the gene is essential, very few transformants were

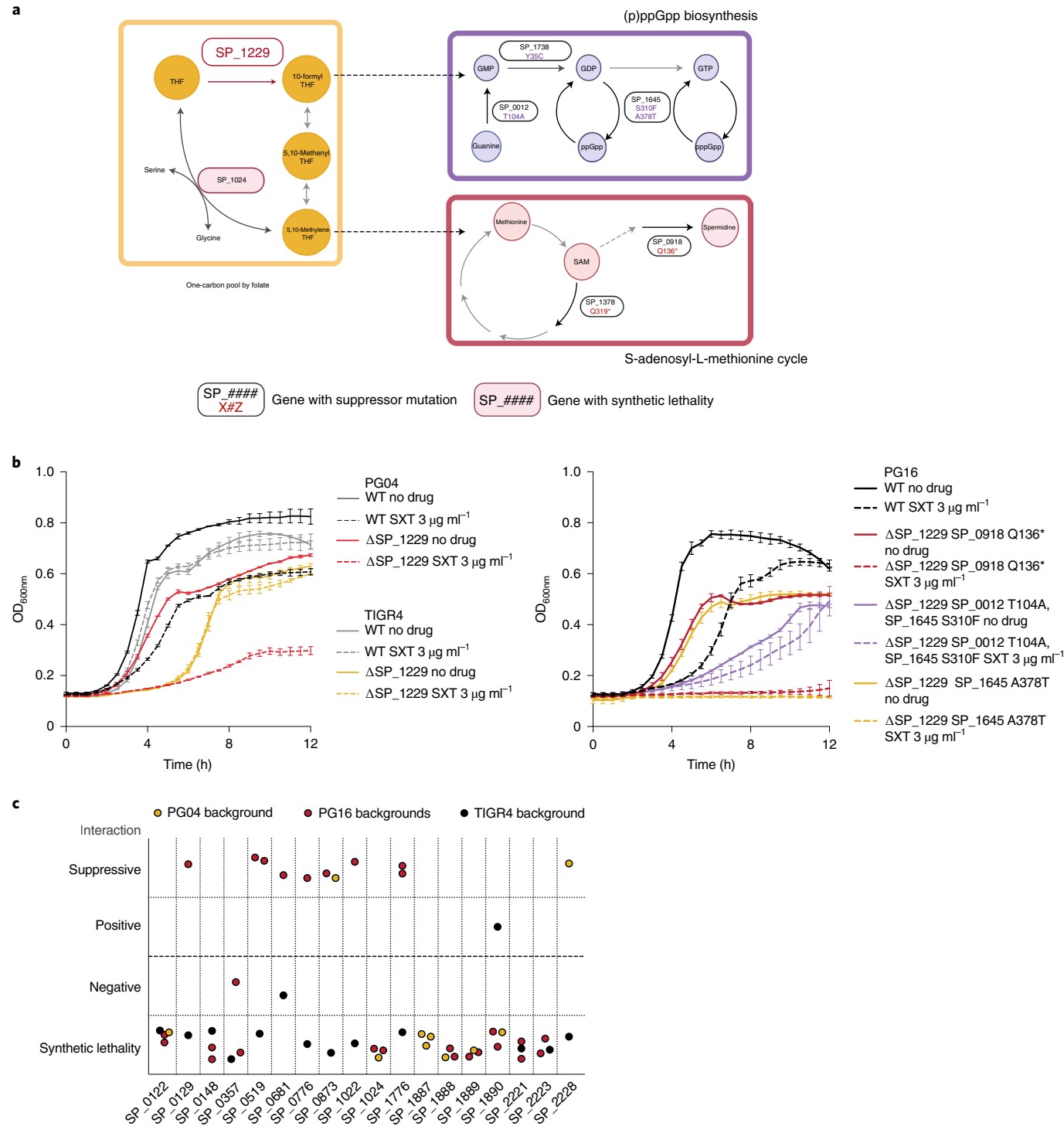

**Fig. 6 | *S. pneumoniae* strains can evolve different mechanisms to overcome the essentiality of formate tetrahydrofolate reductase (SP_1229). a**, Role of the product of the strain-dependent essential gene SP_1229 in the one-carbon pool by folate pathway and suppressor mutations in PG16 bypassing SP_1229 essentiality (boxes with white background). Dashed arrows indicate multiple metabolic reactions (not-shown) that connect the depicted pathways. Two of these mutations, present in two independent backgrounds, create premature stop codons in SP_0918 (Q136*) and SP_1378 (Q319*), which connect to the one-carbon pool by folate pathway through the S-adenosyl-methionine cycle. SP_0918, when deleted in strain D39, diminishes the production of spermidine[72], while SP_1378 codes for an *S*-adenosyl-ʟ-methionine (SAM)-dependent RNA-methyltransferase. Three additional PG16-SP_1229 knockouts contain suppressor mutations in genes involved in guanosine ribonucleotide metabolism and (p)ppGpp biosynthesis (SP_0012-T104A and SP_1645-S310F, SP_1645-A378T and SP_1738-Y35C, respectively). **b**, Growth of WT strains PG04, TIGR4 and PG16, and their SP_1229 knockouts in rich medium in the presence/absence of the tetrahydrofolate (THF) synthesis inhibitory drug SXT. SP_1229 is non-essential in PG04 and TIGR4, and essential in PG16. Data are from $n=3$ biologically independent samples per condition, independent repetition of each experiment (3) showed similar results. Error bars represent s.d. **c**, Identification of genetic interactions through Tn-seq using libraries constructed in two backgrounds where SP_1229 is non-essential (PG04 and TIGR4) and two knockouts derived from PG16 with different suppressor mutations (SP_0918 Q136* and SP_1378 Q319*).

obtained. Excitingly, WGS confirmed that these transformants lack a functional SP_1205 copy while revealing multiple putative suppressive single nucleotide polymorphisms (SNPs) (Supplementary Data 10), suggesting that its essentiality may be bypassed.

Many of the putative bypass mutations and synthetic lethal interactions obtained from Tn-seq in ΔSP_1205-PG06 associate with two cellular processes: (1) the utilization of disaccharides leading to capsule biosynthesis; and (2) the recycling of UP (Fig. 5b). To validate the first process, growth was evaluated with the disaccharide sucrose and minodronate (MNDR), a drug targeting SP_1205 (Extended Data Figs. 6 and 7a). For wild-type strains, MNDR sensitivity increases in the presence of sucrose, highlighting SP_1205's importance, with a disaccharide as the main carbon source. Tn-seq genetic interaction experiments show that the capsule sugar transferase has a robust negative interaction with SP_1205, demonstrating capsule production essentiality in the absence of SP_1205 (Supplementary Data 11). Additionally, growth in the presence of sucrose negatively affects *S. pneumoniae* capsule production[51]. Synthetic lethality between SP_1205 and genes involved in disaccharide utilization can therefore be explained by a consequential decrease in capsule production.

A second set of suppressive SNPs and genetic interactions relate to UP recycling (Fig. 5b, and Supplementary Data 10 and 11). UP homoeostasis depends on a critical balance between demand, synthesis and recycling, in which SP_1205 sits at the root. Bacitracin is a drug that prevents undecaprenyl diphosphate (UPP) recycling to UP, and genes previously described as induced by and required to overcome bacitracin are synthetically lethal with SP_1205[52]. We hypothesized, that ΔSP_1205 would be more sensitive to bacitracin, which we confirmed, indicating that when synthesis is inhibited, recycling of UPP becomes more crucial (Extended Data Fig. 7b). Interestingly, PG06-WT, in which SP_1205 is non-essential, is more sensitive to bacitracin than for instance TIGR4, in which SP_1205 is essential, suggesting that in PG06, UP homoeostasis is skewed towards recycling. Importantly, these data show that *S. pneumoniae* can evolve different solutions to make SP_1205 dispensable. However, besides these solutions, any induced stress that affects UP homoeostasis or capsule production makes SP_1205 essential.

**Fhs essentiality relates to the stringent response.** The enzyme encoded by SP_1229 (Formate tetrahydrofolate ligase, Fhs) belongs to the one-carbon pool by folate pathway. In this pathway, tetrahydrofolate derivatives donate one carbon for the synthesis of purines and amino acids (Fig. 6a)[53]. Knockouts were recovered for two strains where SP_1229 is non-essential, and for PG16 (where it is essential) but only in combination with potential suppressor SNPs (Fig. 6a and Supplementary Data 10).

To investigate how SP_1229 essentiality can be bypassed, we evaluated growth in the presence of cotrimoxazole (SXT) (Fig. 6b), which inhibits synthesis of SP_1229's substrate tetrahydrofolate. PG04 and the PG16-derived knockouts grow slower in rich medium, and except for one of the PG16-derived knockouts have increased sensitivity to SXT. In contrast, the TIGR4-derived knockout grows similarly to the wild type. These results show that besides the strain-dependent essentiality of SP_1229, the effects of deleting the gene differ across strains where the gene is non-essential.

Tn-seq experiments with PG04, TIGR4 and two different PG16-derived knockouts revealed shared and strain-dependent genetic interactions with SP_1229 (Supplementary Data 11). Interestingly, nine genes have an opposite genetic interaction in TIGR4 compared with the other three backgrounds (Fig. 6c). A key interaction exists with the serine hydroxymethyltransferase gene SP_1024. This enzyme is synthetically lethal with SP_1229 in PG04 and PG16 but not in TIGR4, and can bypass SP_1229 by using serine as substrate (Fig. 6a). Two lines of evidence suggest that this bypass mechanism links with (p)ppGpp metabolism. First,

some strains such as PG16 require additional changes to bypass SP_1229, including suppressor SNPs in the (p)ppGpp synthetase enzyme RelA (SP_1645), and two related enzymes SP_0012 and SP_1738. Second, in D39, mupirocin—a drug that robustly triggers the stringent response—induces the expression of SP_1024 in an SP_1645-dependent manner[54]. Overall, these results show that different bypass mechanisms exist to overcome SP_1229 inactivation.

## Discussion

We used a systems biology approach to determine, at the species level, how a bacterial pan-genome might influence gene essentiality, and whether essential genes can evolve to become non-essential. We identified a set of 206 universally essential genes whose essentiality cannot be circumvented and 186 strain-dependent essential genes for whom the genetic background is a significant modulator of essentiality. An advantage of this strain-dependent essentiality is that it enables an approach to infer gene function, for instance, through analysis of compensatory mechanisms. Our main findings extend beyond *S. pneumoniae* biology, with at least two important consequences. First, rather than classifying genes as essential, it may be more useful to approach essentiality in a more quantitative manner[6] that includes context-dependency and evolvability. Second, our findings that many essential genes are influenced by their genetic background may undermine their attractiveness as candidate drug targets. However, we show that elimination of core strain-specific essentials come with a large fitness cost, especially in vivo, indicating that drug inhibitors would still be efficacious. Moreover, our genetic interaction and drug-sensitivity assays highlight that strain-dependent essentials could work in synergistic approaches. Lastly, a pan-genome with strain-dependent essentiality, and more specifically, strains in which essentiality can be altered by changes to the genetic background, may also create new targeted drug screening opportunities. Different sets of strains with changes to different pathways and/or processes which thereby enable differential essentiality can be used to identify compounds that target a specific gene product and/or process. We are currently exploring these and other approaches, which highlights how exploration of basic biology can potentially go hand in hand with the development of new antimicrobial strategies.

## Methods

**Strains and growth conditions.** *S. pneumoniae* strains used in this study are listed in Supplementary Table 1. Thirty of them (PG01 to PG30) belong to a surveillance study done in a Nijmegen hospital, the Netherlands[23], while the other six are lab strains available at our lab.

Different strains were grown on tryptic soy agar or blood agar base no. 2 (Sigma-Aldrich) plates supplemented with 5% defibrinated sheep's blood at 37 °C in a 5% CO$_2$ atmosphere. Liquid cultures were grown statically in THY, C+Y or semi-defined minimal media (SDMM) at pH 7.3, with 5 μl ml$^{-1}$ oxyrase (Oxyrase)[2] at the same incubation conditions as plates. For growth curve assays, strains were grown in THY until an optical density (OD)$_{600}$ of ~0.5, pelleted and resuspended in the same volume of phosphate saline buffer (PBS). OD$_{600}$ was adjusted to 0.05 in PBS and 20 μl of this suspension were diluted in 180 μl of different media conditions in wells of flat-bottom 96-well plates. OD$_{600}$ measurements were taken on a BioSpa 8 plate reader (BioTek) and experiments were repeated at least three times. Different modifications to SDMM were assayed as described in figure legends and text.

**Genomic DNA isolation and PacBio sequencing.** To obtain high-quality and high-yield genomic DNA for PacBio sequencing and Tn-seq libraries construction, strains were grown in 20 ml THY (plus oxyrase, plus catalase 200 units per ml) until an OD$_{600}$ of ~0.7, and cells were recovered by centrifugation (3,000 g, 7 min). Pellets were subsequently processed and genomic DNA purified using QIAGEN Genomic-tip 100/G columns. Initial lysis of cells was achieved by resuspending pellets in a lysis buffer adapted for *S. pneumoniae* (10 mM Tris·Cl, pH 8.0; 10 mM EDTA pH 8.0; 0.1% Tween-20; 1.1% Triton X-100, sodium deoxycholate 1.5%) containing 200 μg ml$^{-1}$ RNAse A (Macherey-Nagel) and incubated for 15 min at 37 °C. The rest of the protocol was performed as described in the QIAGEN gDNA handbook. For mucoid strains, such as PG23 and PG24 (serotype 3), cells were washed once with 1 M NaCl and once with PBS before proceeding with lysis.

SMRT-seq was carried out on a PacBio Sequel-I instrument (Pacific Biosciences). Genomic DNA concentration was determined using the QuBit dSDNA HS (High Sensitivity) assay kit (Thermo Fisher) and purity was calculated using a Nanodrop Spectrophotometer 1000. Genomic DNA samples (3 μg) were sheared to an average size of 10 kb via G-tube (Covaris) before library preparation. Libraries were then generated with SMRTbell Express Template Prep Kit 1.0 and pooled libraries were size selected using the BluePippin system with 0.75% Pippin Gel Cassettes and Marker S1 (Sage Sciences) at a 4 kb minimum threshold. Sequencing reads were processed using the Pacific Biosciences' SMRTAnalysis pipeline version 8.0.0.80529 and assembled using Microbial Assembler.

Closed single contigs were annotated using the NCBI Prokaryotic Genome Annotation Pipeline and genome assemblies and assembly information are made available as part of GenBank Bioproject PRJNA514780.

**Pan-genome core, accessory genome and phylogeny determination.**
A pan-genome study group consisting of 208 isolates was constructed as follows: to the 30 strains selected from the Nijmegen surveillance, six diverse clinical and experimental lab strains were added: TIGR4 (first *S. pneumoniae* sequenced strain isolated from Norway), Taiwan19F (a penicillin-resistant strain isolated from Taiwan), BHN97 (isolated from Norway), two recent US isolates (CT22F and GA22F), and a derived clone of the Griffith-Avery model strain D39. Another 111 strains were added to include diversity obtained from various large-scale survey/sequencing projects that were performed across the world: 30 strains from a set of isolates collected from children during routine visits to primary care physicians in Massachusetts, United States[25], 30 from a refugee camp in Maela, Thailand[24], 30 from Southampton (United Kingdom) ('WGS of carried *S. pneumoniae* during the implementation of pneumococcal conjugate vaccines in the UK', Bioproject PRJEB2417) and 21 from the Malawi-Liverpool-Wellcome Trust Clinical Research Programme (MLW) strain archive[26]. Furthermore, 23 *S. pneumoniae* closed genomes available in RefSeq[55] and 38 strains isolated before 1974[27] completed the 208 strains set. Similar numbers of strains were chosen from each project to obtain a well-balanced dataset that is not dominated by a single study.

Three pipelines for orthologous gene clustering and pan-genome analysis (PanX[28], PPanGGOLin[29] and BF-Clust[30]) were applied to a study group of 208 sequenced isolates including the 36 strains of our collection (Supplementary Table 3 and 4). In the case of BF-Clust methodology, clusters were subclustered according to their genomic neighbourhood to discriminate between paralogues (Supplementary Table 5). When referring to specific genes, we used as a convention the old locus tag from TIGR4 (SP_#). For those genes not present in the TIGR4 genome, we used the D39 locus tag (SPD_#), Taiwan19F (SPT_#), or the BF-Clust subcluster ID (#_#).

Average nucleotide identity values between PG-collection strains were determined using KBase platform[56]. The phylogenetic tree of Extended Data Fig. 1a was obtained as an output from the PanX pipeline, constructed on the basis of the core genes SNPs of the selected 208 strains and edited using iTol[57].

**Tn-seq libraries construction.** Six independent transposon libraries, each containing 10,000 to 20,000 insertion mutants, were constructed with transposon Magellan6 in the different strains as described previously[34,58], with the following modifications: (1) gDNA used was isolated as described for PacBio sequencing, (2) Marc9 transposase expression was induced with 0.5 mM Isopropyl β-D-1-thiogalactopyranoside (IPTG) in 2YT medium with 2% ethanol at 25 °C during 4 h and (3) transformation reactions were scaled up to 4 ml volume. Transposon mutants were recovered in blood agar base no. 2 plates supplemented with 5% sheep's blood with 200 μg ml⁻¹ spectinomycin. Libraries stock cultures were grown several independent times in THY medium, gDNA was isolated using Nucleospin tissue kit (Macherey-Nagel), and Tn-seq sample preparation and Illumina sequencing were done as previously described[34,58].

**Tn-seq essential genes, fitness and genetic interactions determination.** The binomial method of the TRANSIT package was used for determination of essential genes in the 22 strains with constructed libraries[59]. Binomial categorization relies on the calculation of a $z$-value determined by library saturation, the number of teichoic acid (TA) sites in the gene, and a calculated false discovery rate (FDR) of 5% to set the 'non-essential' and 'essential' $z$-value thresholds. 'Uncertain' predicted genes have $z$-values between these thresholds. Ten percent of the sequence of a gene from the 5′ and 3′ ends was omitted for the calculation of the $z$-value. To compare between strains and determine the essential genes classes, only strains with a saturation higher than 35% were considered (17 strains). The maximum and minimum $z$-values observed for each gene across strains were calculated. All genes with a maximum $z$-value above the essential threshold were included as part of the essentialome. Core genes with a minimum $z$-value above the essential threshold were classified as universal essentials. Core genes with maximum $z$-values above the essential threshold (thus part of the essentialome) but minimum $z$-values below the essential threshold were classified as core strain-dependent essential genes. One caveat to consider about this last classification is that for some genes, minimum $z$-values are in the uncertain zone (between essential and non-essential thresholds). This could lead to some false positives, especially for those genes with minimum $z$-values close to the essential threshold.

To obtain the average mutant fitness ($\overline{W}$) in SDMM medium, Tn-seq libraries were first grown in THY until an $OD_{600}$ of ~0.5, pelleted and diluted to an $OD_{600}$ of 0.1 in SDMM, with 20 mM glucose as carbon source. Cells were grown for 4 generations (~3 h). Genomic DNA from THY and SDMM-grown cells were processed for Tn-seq. $W_i$ (where i denotes a specific insertion mutant) for each transposon insertion was calculated as previously described by comparing the fold expansion of the mutant relative to the rest of the population at the initial time (THY) and the final time (SDMM)[33,58,60]. All the mutants $W_i$ in a specific gene, excluding those with insertions in the first and last 10% of the gene total sequence, were then used to calculate the $\overline{W}$ and standard deviation for the gene in question. To statistically determine whether a gene interruption affected growth in SDMM, the following analysis for each strain was performed: (1) essential genes specific for each strain and genes with less than three data points were excluded from the analysis, (2) median average fitness across all genes was calculated (on the basis of results from Tn-seq experiments with high bottleneck, we assumed that the median is the better estimate for a fitness value that represents no effect on growth), (3) the observed $\overline{W}$ and the median (expected W) were compared by a one sample $t$-test, and obtained $P$ values were corrected for multiple comparisons by an FDR of 5% using the two-stage step-up method[61] and (4) genes tagged as a true discovery and with a $\Delta W$ ($\overline{W}$ observed − W expected, median) absolute value higher than 0.2 were considered required ($\Delta W < -0.2$) or disadvantageous ($\Delta W > 0.2$) for growth in SDMM.

Genetic interactions were determined by comparing Tn-seq binomial predictions (suppressive and synthetic lethality interactions) or SDMM fitness calculations (negative and positive interactions) between libraries constructed in wild-type backgrounds and specific knockout backgrounds (Extended Data Fig. 3c). Genes that changed their categorization from 'essential' in the wild-type (WT) context to 'non-essential' in the knockout context represent suppressive interactions, while genes that changed the opposite way represent synthetic lethality. To reduce false negatives, genes that changed their categorization from 'uncertain' in WT to 'essential' in knockout backgrounds were also tagged as synthetic lethality. These last synthetic lethality interactions thus have a lower confidence, especially for those 'uncertain' genes with binomial $z$-values closer to the 'essential' threshold. Positive and negative interactions were determined as follows: (1) SDMM growth rate ratio between site-directed constructed knockouts and WT was used as multiplicative model correction factor, (2) each mutant $\overline{W}$ obtained from libraries constructed in the knockout and WT were multiplied by the correction factor, (3) the corrected $\overline{W}$s were compared by a two sample $t$-test, and obtained $P$ values were corrected for multiple comparisons by an FDR of 5% using the two-stage step-up method of Benjamini and Hochberg[62], (4) genes tagged as a true discovery and with a $\Delta W$ ($\overline{W}$ knockout corrected − $\overline{W}$ wild type corrected) absolute value higher than 0.2 were considered to have a negative ($\Delta W < -0.2$) or positive ($\Delta W > 0.2$) interaction with the respective knocked-out gene.

**Transcriptome.** Cells were grown in triplicate in THY medium until an $OD_{600}$ of ~0.3. Then, RNA isolation, sample preparation and Illumina sequencing were performed as previously described[63]. Raw data analysis was also performed as described in ref. [63], with the exception that aggregated feature counts were transcripts per million normalized to compare across the different genes[64].

**Site-directed gene deletions and whole-genome sequence analysis.** Site-directed gene knockouts were constructed by replacing target gene sequences with a chloramphenicol and/or spectinomycin resistance cassette as described previously[34], but using up to 1,000 ng of the allele exchange PCR product. All PCR reactions were done using Q5 polymerase (NEB). Primers used and knockouts constructed are described in Supplementary Data 12 and 13. Randomly selected colonies and in some cases, whole transformation populations, were recovered and their gDNA isolated using Nucleospin tissue kit (Macherey-Nagel). Deletions were confirmed by two PCR reactions using primers outside of the PCR product and primers specific to the antibiotic marker used (F0-MR, R0-MF, Extended Data Fig. 3a). Products of around 2 kb were expected for the two PCR reactions. The amplification of a product of around 4 kb in one of the reactions indicates that the clone or population consisted of merodiploids.

Complementation of SP_1603 knockouts was achieved by cloning a WT copy of SP_1603 gene under the control of the constitutive promoter P3 at the non-coding region downstream of SP_1885 (CEP locus)[65].

For identification of putative suppressive mutations, strain gDNA concentrations were measured on a Qubit 3.0 fluorometer (Invitrogen) and diluted to 10 ng ul⁻¹ in 5 mM Tris HCl for library preparation using a Nextera kit (Illumina). Libraries were sequenced on an Illumina NextSeq500 and reads were mapped to their corresponding reference genomes. Single nucleotide mutations, deletions and new arrangements were identified using the breseq pipeline[66]. Merodiploids were distinguished from knockouts by PCR as described above or by breseq. To distinguish merodiploids from knockouts, sequencing coverage of the target gene and the homology arms in the 3 kb product (Extended Data Fig. 3b) were examined. In a merodiploid, coverage of left and right arms is about double of the average coverage due to duplication, while the coverage of the target gene remains the same as the average. Another characteristic of merodiploids

identifiable by breseq is that the left and right homology arms are jointed with a frequency of around 50%.

Some suppressive mutations were validated by co-transforming WT strains with a spectinomycin cassette that inserts in the neutral region between genes SP_2105-SP_2106 and a PCR product with the SNPs to validate in a 1:10 ratio. It is expected that about 20% of spectinomycin-resistant colonies obtained should contain the SNP[67]. The whole population was pooled and transformed with the target essential suppressed gene.

**Membrane polar lipid composition determination.** Strains were grown in C+Y medium at 37 °C to an $OD_{600}$ of 0.2, and 50 μCi of (1-$^{14}$C)acetate was added. After a 2 h labelling period, cells were collected and washed twice with PBS. The lipids were extracted by the Bligh and Dyer method[68], and total incorporation was quantified by scintillation counting using Tri-Carb 2910 TR (Perkin Elmer). The lipids were separated by loading equivalent amounts of radioactivity on silica gel 60A thin-layer plates (Partisil LK6D; Whatman) and developed in chloroform:methanol:acetic acid:water (80:10:14:3, v/v/v/v). After drying, the thin-layer plate was exposed to a phosphorimaging screen overnight. The signal intensity was read using a Typhoon FLA 9500 (GE Healthcare Life Sciences) phosphorimager and the distribution of label quantified using ImageQuant TL (GE Healthcare Life Sciences). The results are from two independent experiments.

**Data visualization and statistics.** Figure panels were created with Biorender.com (Boston College full license) and GraphPad Prism 9. CorA protein structures were modelled with SWISS-MODEL[69] and edited using Chimera.X.1.2.5. Statistical analyses were performed using GraphPad 9 and R 1.4.1106.

**Reporting summary.** Further information on research design is available in the Nature Research Reporting Summary linked to this article.

## Data availability

Genome assemblies, assembly information, Tn-seq, RNA-seq and WGS raw data are available as part of Bioproject PRJNA514780 (https://www.ncbi.nlm.nih.gov/bioproject/PRJNA514780). Source data are provided with this paper.

## Code availability

Tn-seq, RNA-seq and WGS data were analysed using our in-house developed Aerobio platform 2.3.0 available at https://github.com/jsa-aerial/aerobio (Anthony, J. S. & van Opijnen, T. A. DAG computation server for fully integrated and automated massively parallel sequencing analyses. 2019). BF-Clust, MAP files preparation for TRANSIT analysis and Tn-seq statistical testing codes can be found in the fork repositories at https://github.com/frosconi.

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

## Acknowledgements

DNA sequencing was performed at the Boston College Sequencing Core. We thank the invaluable contributions of A. Cremers, for sharing the strains isolated in Nijmegen; J. Radivojac and S. Cotton for help in RNA-seq and PacBio sequencing experiments; Z. Zhu, B. Sundaresh and S. Espinoza for valuable discussions. This work was supported by a PEW Latin America Fellowship and Charles King Trust Fellowship to F.R.; NIH National Institute of Allergy and Infectious Diseases grants R21 AI117247 and R01 AI110724 to T.v.O., and grant U01 AI124302 to T.v.O. and J.W.R.; and NIH National Institute of Dental and Craniofacial Research grant R01DE027850 to C.D.J.

## Author contributions

F.R and T.v.O. devised the study and wrote the manuscript. F.R., E.R., J.L. and M.F. performed wet-lab experiments and data collection. D.S., J.L. and D.S.J. performed pan-genome orthologue cluster analysis. J.A. and D.S. contributed to the Tn-seq dry-lab analysis pipeline. J.W.R. and C.R. contributed to key conceptual ideas. C.D.J. performed PacBio sequencing analysis. F.R and T.v.O performed data analysis and interpretation. All authors contributed to manuscript editing and approved the final paper.

## Competing interests

The authors declare no competing interests.

## Additional information

**Extended data** is available for this paper at https://doi.org/10.1038/s41564-022-01208-7.

**Correspondence and requests for materials** should be addressed to Tim van Opijnen.

a

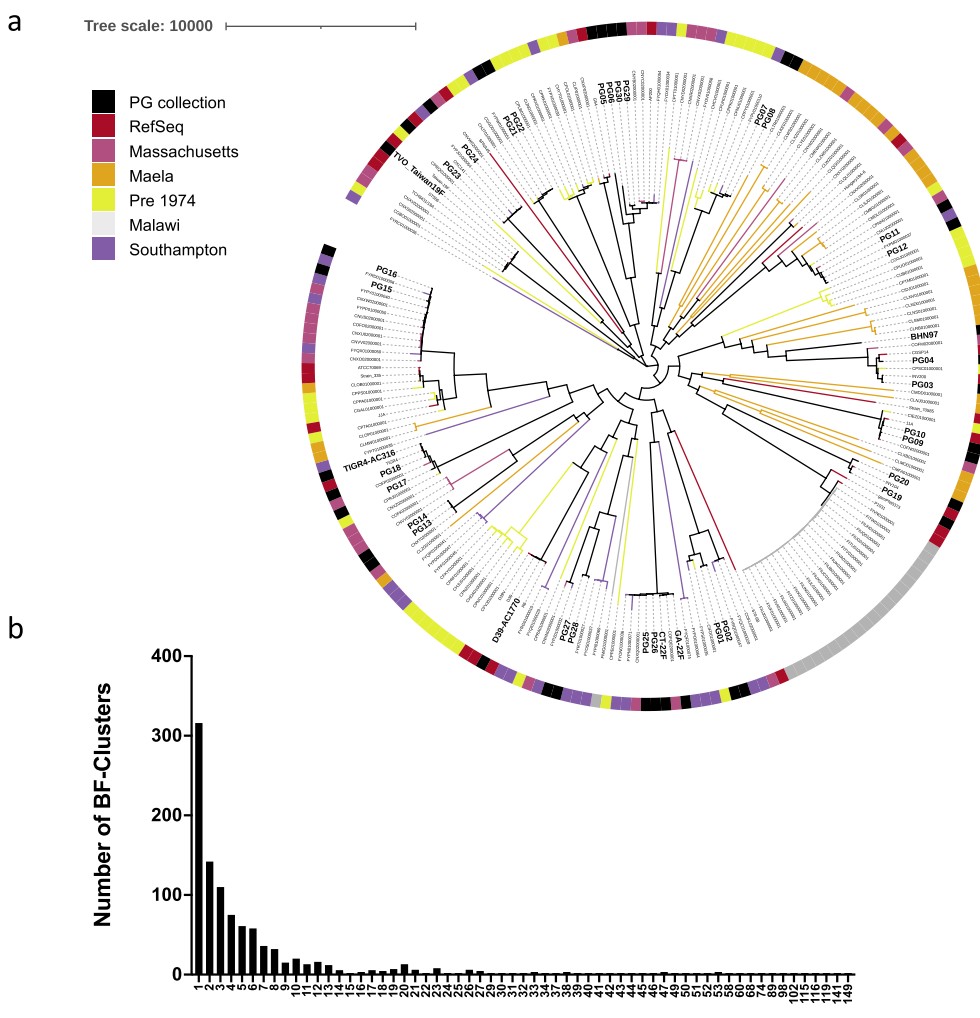

b

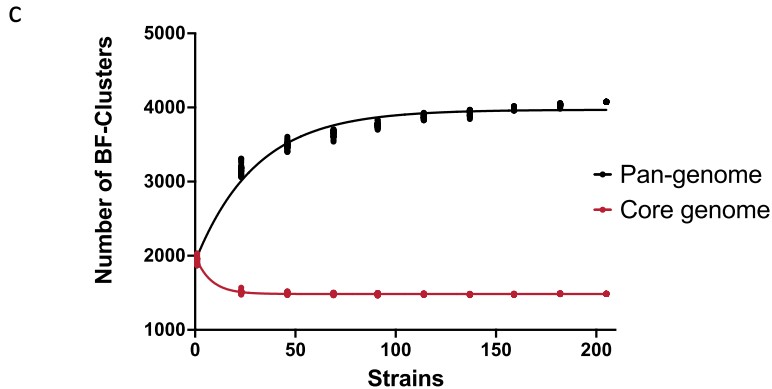

c

**Extended Data Fig. 1 |** *Streptococcus pneumoniae* P̲an-G̲enome collection phylogenetic diversity and pan-genome coverage. a) Maximum-likelihood phylogenetic tree based on core genes SNPs of the 208 strains in the pan-genome study group. This group consists of the 36-strain PG-collection, 23 reference genomes, 111 strains from several worldwide surveillances' programs (30 from Massachusetts, US, 30 from Maela, Thailand, 30 from Southampton, UK, and 21 from Malawi), and 38 isolates obtained before 1974. Labels from the strains selected for our PG collection are in bold and have a larger font size. Black outer strip color symbol and black tree branches also indicate the PG collection strains. Panel shows that PG-collection scatters evenly through the phylogenetic tree. b) Number of BF-clusters not-represented in the 36 strain PG-collection. More than half (55%) of the non-represented genes are present in less than 5 isolates. c) Pan-genome content saturation.

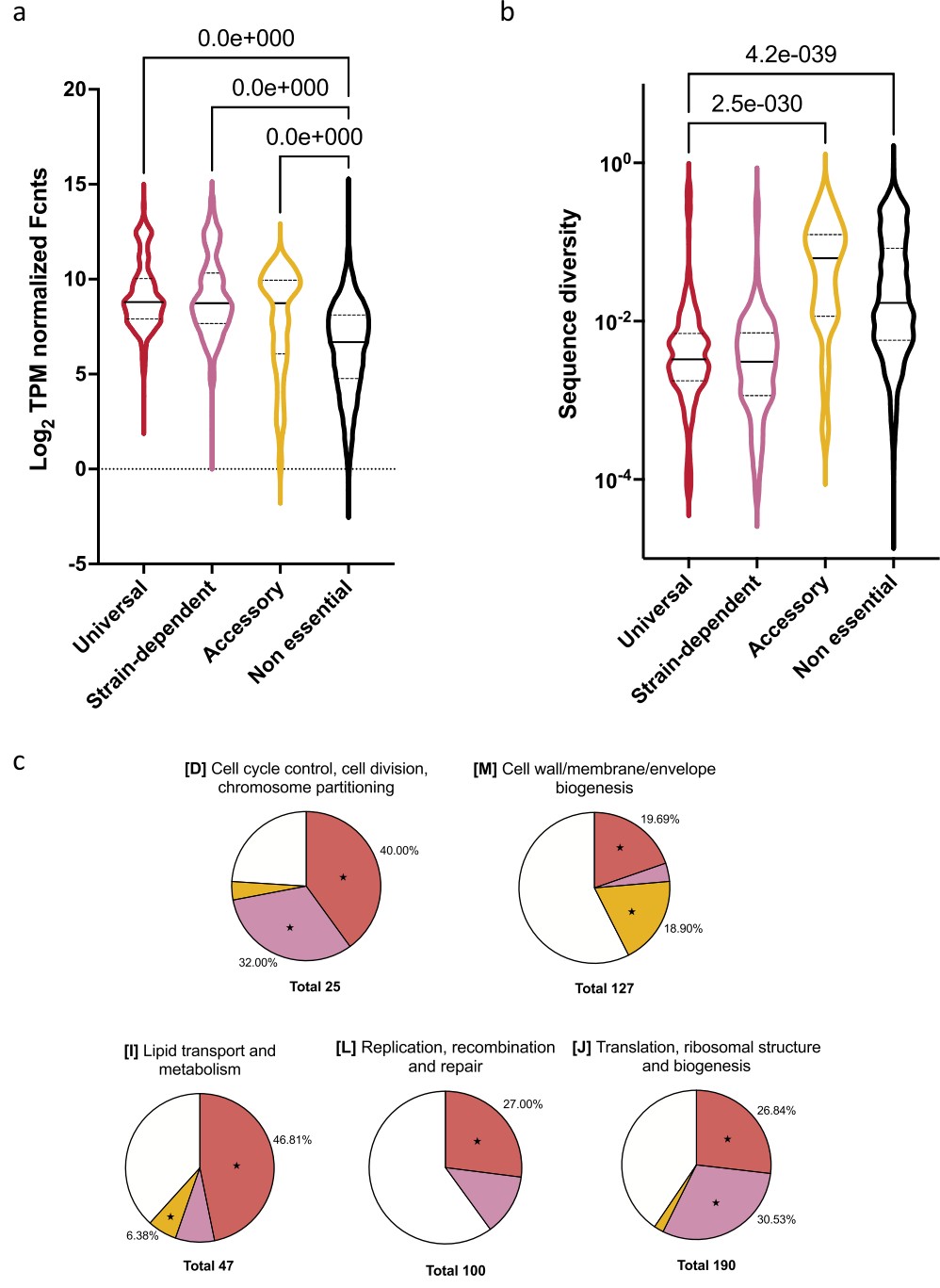

**Extended Data Fig. 2 | See next page for caption.**

**Extended Data Fig. 2 | Characteristics of the different essentialome classes.** a) Gene transcript abundance obtained by RNAseq collected from early exponential phase cultures of the 36 PG-collection strains growing in rich THY medium. Graph shows genes grouped by their essentialome classification. Black lines inside each plot represent the median, and dotted lines the first and third quartiles. $p$-values are obtained from a Tukey's corrected one-way ANOVA. Results show that core universal, core strain-dependent and accessory essential genes tend to be more highly expressed compared to non-essential genes; b) Sequence diversity of *S. pneumoniae* genes split into the different essentialome classes and measured using the diameters of the clusters obtained by BF-Clust. Black lines inside each plot represent the median, and dotted lines the first and third quartiles. $p$-values are obtained from a Kruskal-Wallis test comparison. Strain-dependent versus accessory and non-essential classes presented the same $p$-value ($<0.001$). Results show that core universal and core strain-dependent essential genes are less genetically diverse than accessory and non-essential genes. The higher diversity in accessory essentials is at least partially caused by many of them being involved in 'addiction-like' systems, for example, systems consisting of two or more genes in which one of the genes is only required if the other/s are present (for example phage repressors, antitoxins from toxin-antitoxin systems, and methylases from restriction-modification systems). These genes may thus tolerate more mutations than a core genome gene involved in central processes, such as DNA replication; c) Functional categories enriched in Universal, Core Strain-dependent, and Accessory essentials. Each chart represents a COG category and the fraction of genes belonging to each essentialome class. Color coding is the same as previous panels. Asterisks indicate enrichments of essentialome class within COG category (adjusted $p$-value$< 0.05$). Analysis shows that universal essential genes are enriched in 5 central cellular processes categories, while core strain-dependent and accessory essentials are enriched for only two of these categories. The most enriched functional category of accessory essentials (cell wall, membrane and envelope biogenesis) can be explained by the high diversity of essential capsule type-specific biosynthesis genes.

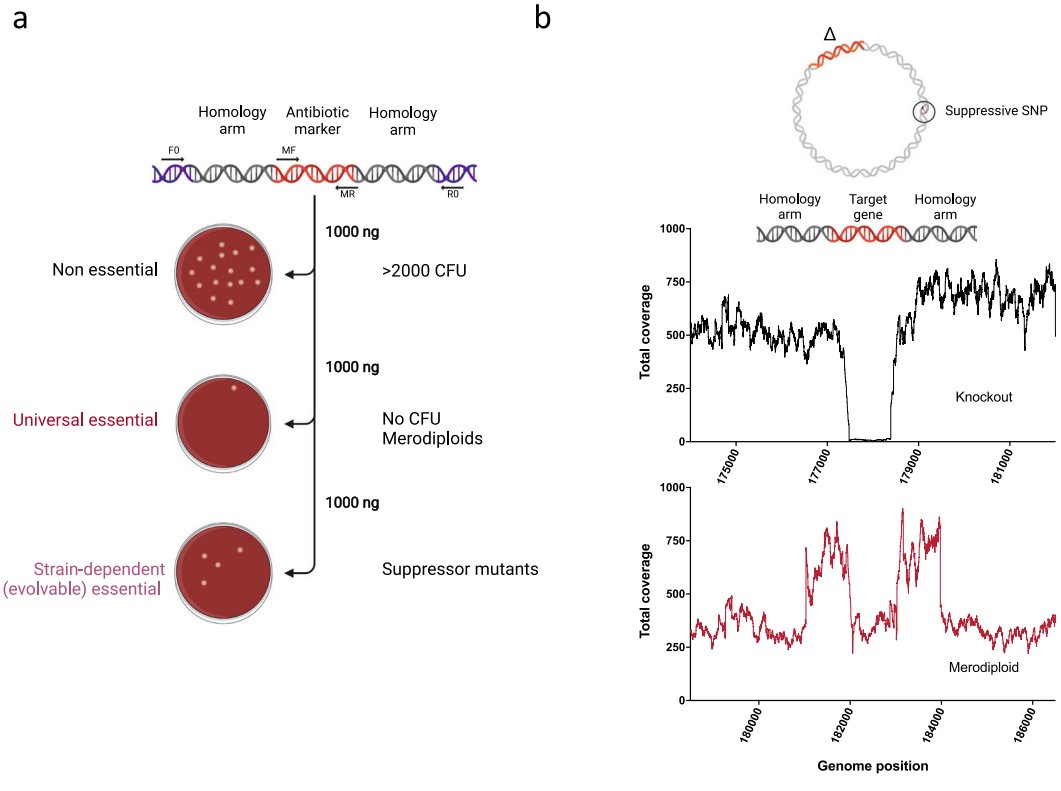

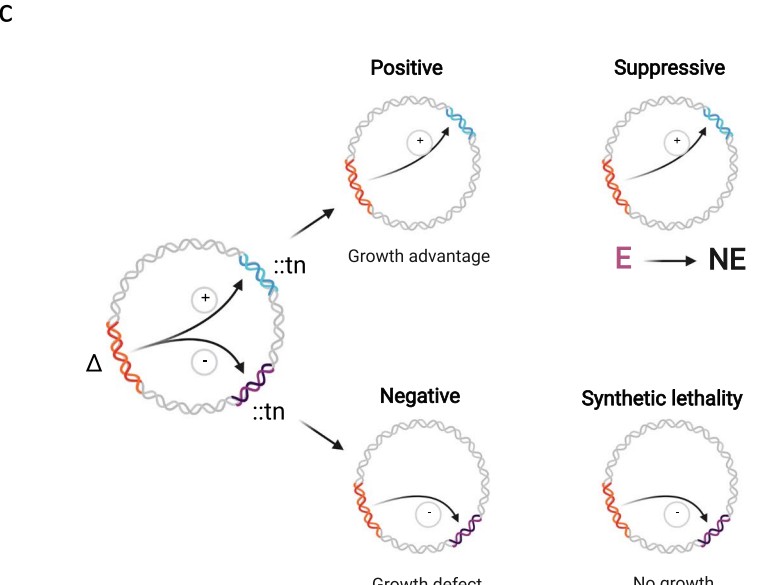

**Extended Data Fig. 3 | Methodology for essentiality validation, and evolvability, and genetic interaction identification of essential genes.** a) Strategy deployed to obtain knockouts in different genes and representative results for each essentialome class. The amount of PCR knockout product used (1000 ng) was 20 times higher than routinely used. b) Knockout confirmation and identification of suppressor mutations. WGS analysis with *breseq* of clones selected from transformations revealed whether clones were true knockouts (targeted gene coverage is near zero, middle panel) or merodiploids (target gene coverage same as average, homology arms coverage twice the average, bottom panel). WGS and *breseq* were also used to identify SNPs in genes putatively involved in gene essentiality suppressor mechanisms. c) Genetic interactions between strain-dependent and non-essential genes. Tn-Seq experiments using libraries constructed in different knockouts identified four different types of interactions: 1) a positive interaction occurs when a double mutant (knockout, "Δ", plus transposon, "::tn") presents a growth advantage compared to the individual mutants; 2) a suppressive interaction occurs when an essential gene becomes non-essential in the context of the knockout; 3) in a negative interaction the double mutant presents a growth defect, and 4) in a synthetic lethal interaction the double mutant is non-viable. Tn-Seq fitness calculations were used to identify positive and negative interactions, and combined with the TRANSIT binomial method to identify suppressive interactions and synthetic lethality.

a

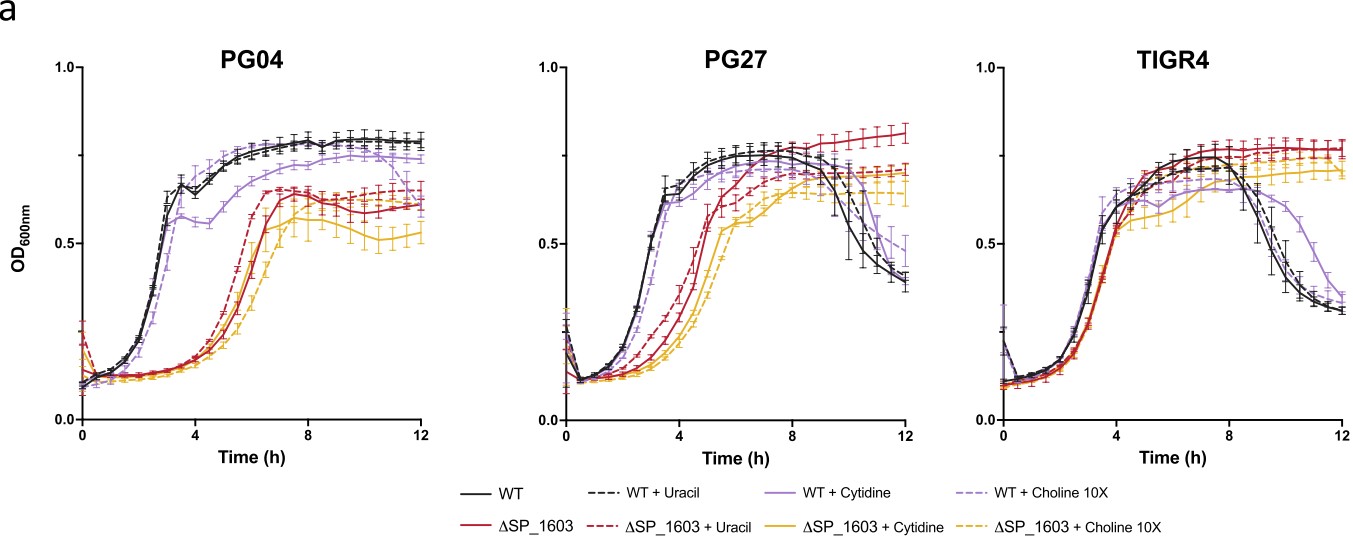

b

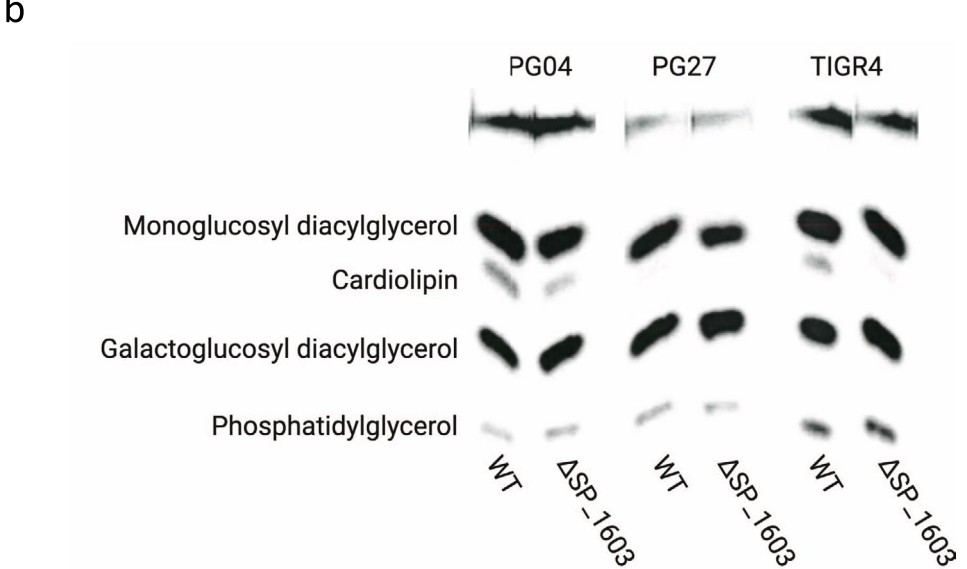

**Extended Data Fig. 4 | SP_1603 knockout characterization.** a) Growth phenotype of wild type and derived SP_1603 knockouts in rich medium (SDMM) in the presence/absence of additional uracil, cytidine, or choline. Knockout phenotypes are potentially caused by a decreased replication rate provoked by a dCTP shortage. While addition of excess uracil and/or cytidine could potentially restore growth, it had no effect. Alternatively, CMP accumulation could be negatively affecting pathways or reactions in which CMP is a product. In *S. pneumoniae*, genes SP_1273 and SP_1274 generate CMP from CDP-choline and are involved in decorating teichoic acid with choline[73,74]. While CMP accumulation could thus affect teichoic acid choline decoration, the addition of 10x more choline to the medium also did not compensate growth. Data are obtained from n = 3 biologically independent samples per condition, independent repetition (3) showed similar results. Error bars represent standard deviation. b) TLC run of C-14 acetate labeled lipids extracted from the different strains. Figure 4c summarizes this experiment.

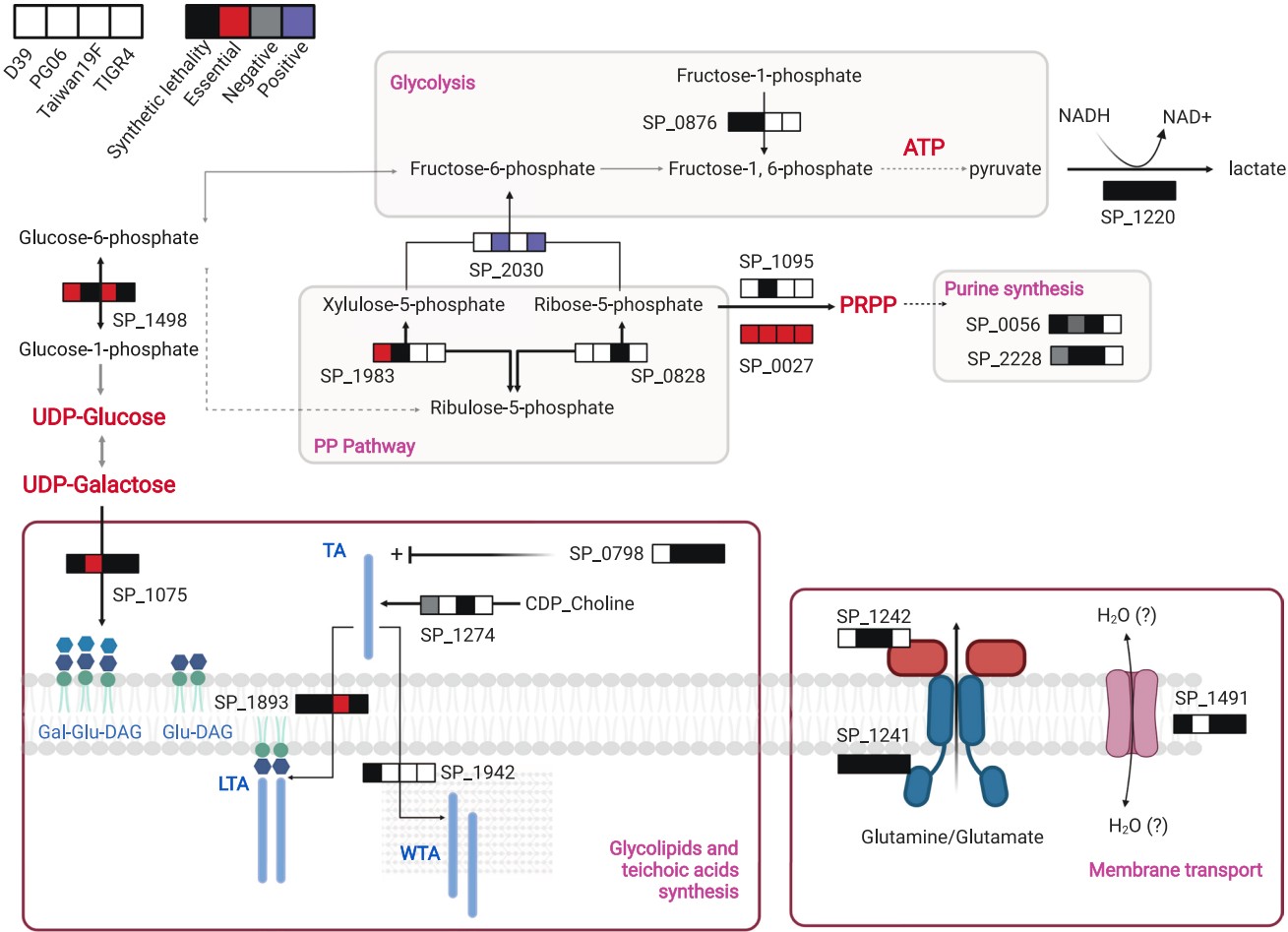

**Extended Data Fig. 5 | Genes and pathways with genetic interactions with capsule sugar transferase genes.** Selection of genetic interactions identified by Tn-Seq using libraries constructed in sugar transferase knockouts derived from four different strains. The order of the bars consisting of 4 squares indicate the strain, and the color the type of interaction that was identified. White square depicts no identified interaction. Metabolites in red bold indicates those that accumulate in acapsular *S. pneumoniae* strains[51]. Gal-Glu-DAG: galactosyl-glucosyl-diacylglycerol, Glu-DAG: glucosyl-diacylglycerol, TA: teichoic acid, LTA: lipoteichoic acid, WTA: wall teichoic acid.

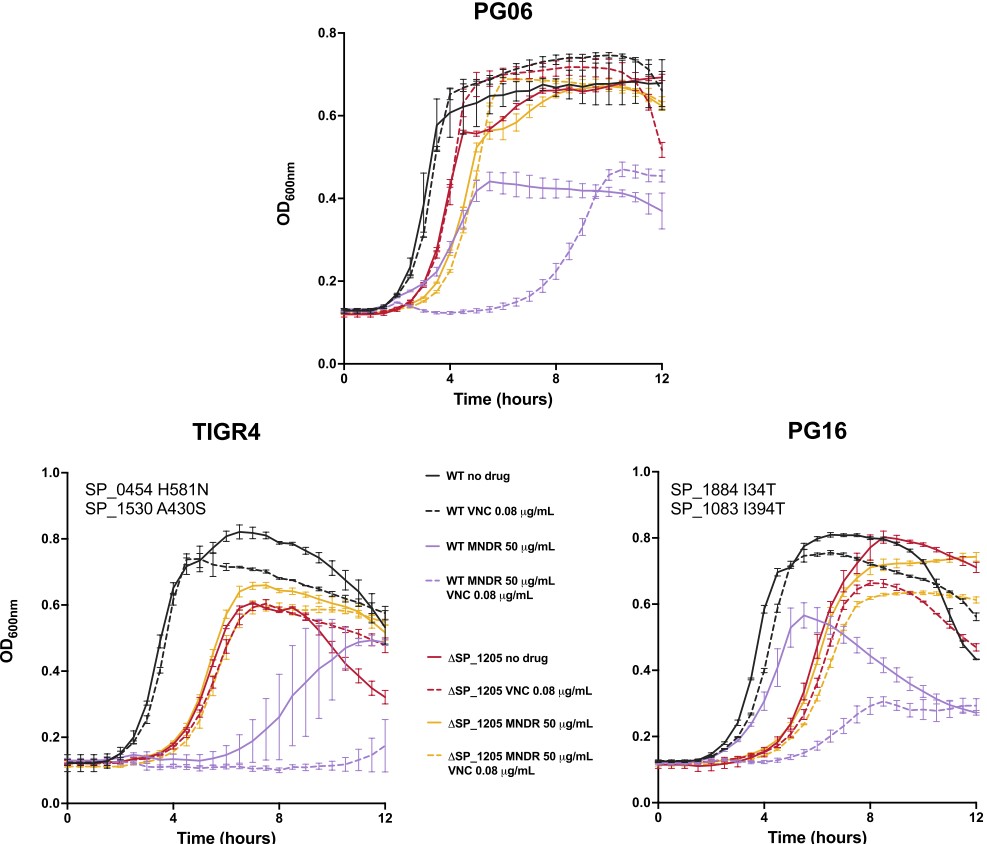

**Extended Data Fig. 6 | Minodronate (MNDR) specifically targets _S. pneumoniae_ farnesyl-PP synthase (SP_1205) and synergizes with vancomycin.**
Growth of WT strains PG06 (in which SP_1205 is non-essential), TIGR4 and PG16 (in which SP_1205 is essential) and their derived SP_1205 knockouts in rich medium (SDMM), in the presence of non-inhibitory concentrations of vancomycin (VNC), in the presence or absence of the SP_1205 targeting drug MNDR, or both. Panels show the suppressor mutations that enable SP_1205 knockout in strains it is essential in. MNDR is a bisphosphonate drug currently in a phase-3 trial for human use, which targets human FPP synthase to suppress bone resorption and bone loss. However, there is evidence that bisphosphonates may also have antimicrobial potential[75]. The drug inhibits growth of the wild-type strains but not the SP_1205 knockouts. Because UP is critical for the cell wall, inhibition of UP synthesis may thus synergize with cell-wall synthesis inhibitors, like vancomycin. The combination of MNDR with vancomycin at a subinhibitory concentration has a synergistic effect on wild-type strains while there is no effect in ΔSP_1205 strains. Results shown are from n = 3 biologically independent samples per condition, independent repetition of each experiment (3) showed similar results. Error bars represent standard deviation.

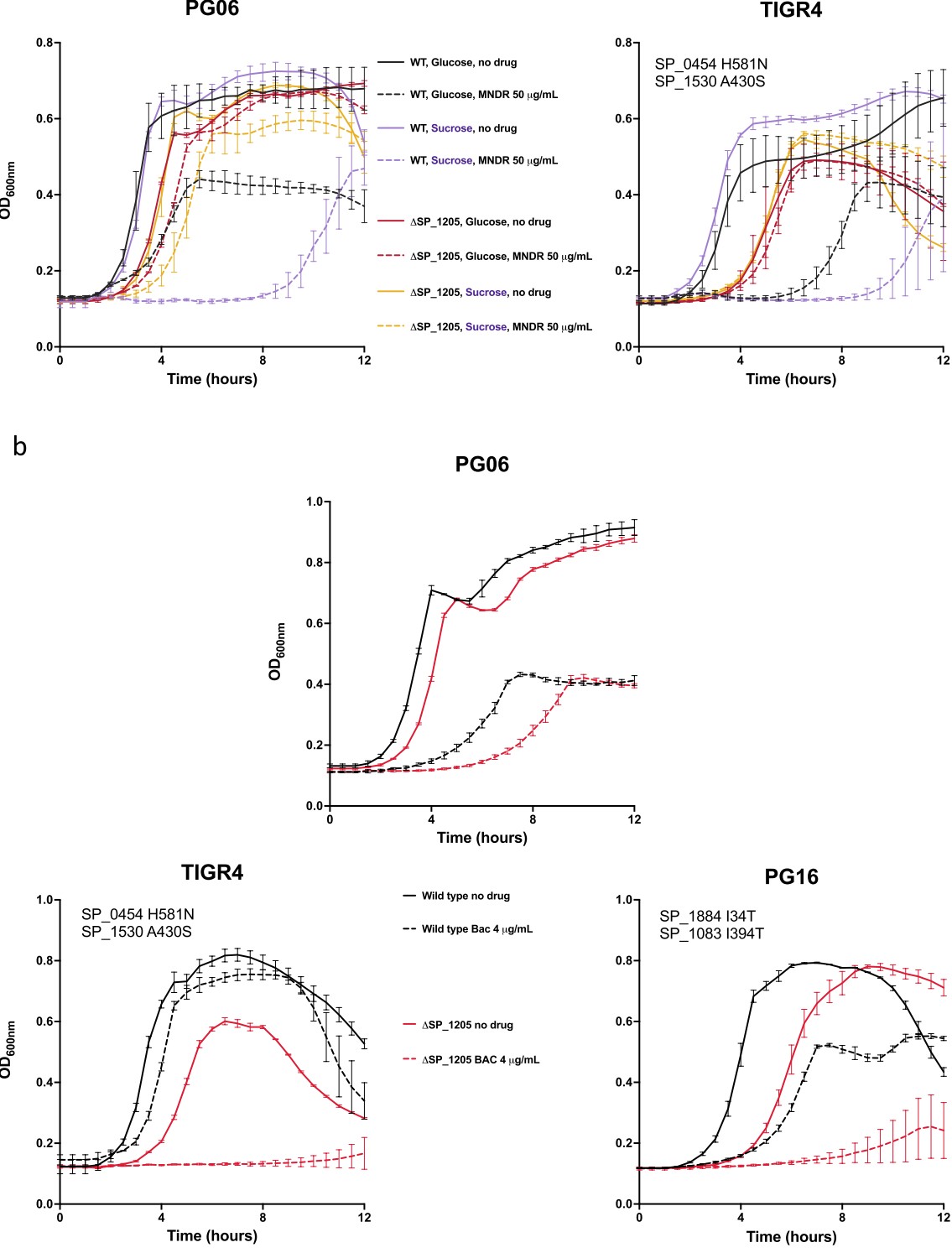

**Extended Data Fig. 7 | *S. pneumoniae* requires SP_1205 when growing in the presence of sucrose or bacitracin.** a) Carbon source and SP_1205 requirement. Growth of WT strains PG06 (in which SP_1205 is non-essential) and TIGR4 (in which SP_1205 is essential) and their derived SP_1205 knockouts in rich medium (SDMM) with either glucose or sucrose as the carbon source, and with or without MNDR. b) Bacitracin sensitivity of SP_1205 knockouts. Growth of WT strains PG06 (in which SP_1205 is non-essential), TIGR4 and PG16 (in which SP_1205 is essential), and their derived SP_1205 knockouts in rich medium (SDMM) in the presence or absence of the UPP recycling inhibitory drug bacitracin (Bac). Data in a and b are the results from n=3 biologically independent samples per condition, independent repetition of each experiment (3) showed similar results. Error bars represent standard deviation.

# Reporting Summary

## Statistics

For all statistical analyses, confirm that the following items are present in the figure legend, table legend, main text, or Methods section.

| n/a | Confirmed | |
|---|---|---|
| ☐ | ☒ | The exact sample size ($n$) for each experimental group/condition, given as a discrete number and unit of measurement |
| ☐ | ☒ | A statement on whether measurements were taken from distinct samples or whether the same sample was measured repeatedly |
| ☐ | ☒ | The statistical test(s) used AND whether they are one- or two-sided *Only common tests should be described solely by name; describe more complex techniques in the Methods section.* |
| ☒ | ☐ | A description of all covariates tested |
| ☐ | ☒ | A description of any assumptions or corrections, such as tests of normality and adjustment for multiple comparisons |
| ☐ | ☒ | A full description of the statistical parameters including central tendency (e.g. means) or other basic estimates (e.g. regression coefficient) AND variation (e.g. standard deviation) or associated estimates of uncertainty (e.g. confidence intervals) |
| ☐ | ☒ | For null hypothesis testing, the test statistic (e.g. $F$, $t$, $r$) with confidence intervals, effect sizes, degrees of freedom and $P$ value noted *Give P values as exact values whenever suitable.* |
| ☒ | ☐ | For Bayesian analysis, information on the choice of priors and Markov chain Monte Carlo settings |
| ☒ | ☐ | For hierarchical and complex designs, identification of the appropriate level for tests and full reporting of outcomes |
| ☒ | ☐ | Estimates of effect sizes (e.g. Cohen's $d$, Pearson's $r$), indicating how they were calculated |

*Our web collection on statistics for biologists contains articles on many of the points above.*

## Software and code

Policy information about availability of computer code

| Data collection | Tn-Seq, RNA-seq and WGS experiments data was obtained using the Nextera Illumina platform and analyzed using the aerobio pipeline (https://github.com/jsa-aerial/aerobio). |
|---|---|
| Data analysis | For analysis of the data we used Aerobio 2.3.0 (https://github.com/jsa-aerial/aerobio), R Studio 1.4.1106, Chimera.X.1.2.5, GraphPad 9, SMRTAnalysis pipeline version 8.0.0.80529. PanX analysis results, MAP files prepping for TRANSIT analysis, and Tn-Seq statistical testing codes can be found in the fork repositories at https://github.com/frosconi. |

For manuscripts utilizing custom algorithms or software that are central to the research but not yet described in published literature, software must be made available to editors and reviewers. We strongly encourage code deposition in a community repository (e.g. GitHub). See the Nature Portfolio guidelines for submitting code & software for further information.

## Data

Policy information about availability of data

All manuscripts must include a data availability statement. This statement should provide the following information, where applicable:
- Accession codes, unique identifiers, or web links for publicly available datasets
- A description of any restrictions on data availability
- For clinical datasets or third party data, please ensure that the statement adheres to our policy

Genome assemblies, assembly information, Tn-Seq, RNA-Seq and WGS raw data are available as part of the Bioproject PRJNA514780 (https://www.ncbi.nlm.nih.gov/bioproject/PRJNA514780). Accesion codes for GenBank biosamples (SAM###) are indicated in Supplementary Data 1.

# Field-specific reporting

Please select the one below that is the best fit for your research. If you are not sure, read the appropriate sections before making your selection.

☒ Life sciences ☐ Behavioural & social sciences ☐ Ecological, evolutionary & environmental sciences

For a reference copy of the document with all sections, see nature.com/documents/nr-reporting-summary-flat.pdf

# Life sciences study design

All studies must disclose on these points even when the disclosure is negative.

| Sample size | The primary dataset of this study (208 strains) was chosen based on geography and date of isolation. The 36 strains of the pan-genome collection were selected based on phylogeny diversity. This collection covers 68% of the pan-genome, and is a number suitable to handle for bench experiments. Sample sizes for Tn-Seq consist of 6 independent libraries for each strain with 10,000 to 20,000 mutants per library. Tn-Seq power increases if different libraries with different mutants frequencies are used. From our previous works, we know 6 libraries are sufficient to obtain confident Tn-Seq results. Sample sizes of 3 were performed for growth assays and all other experiments. The high reproducibility observed among the different replicates proved the number of replicates to be sufficient. |
|---|---|
| Data exclusions | For determination of the essentialome by comparison of the different strains Tn-Seq results, libraries with less than 35% of saturation were excluded. For TPM normalization of RNAseq results, genes with 0 feature counts in at least one replicate were excluded. Genes without assigned clusters (mostly mobile elements, pseudogenes) are not part of the gene clusters analysis. |
| Replication | For Tn-Seq six independent libraries were constructed for each strain. RNA-seq experiments consisted of three independent replicates. Knockout validations consisted in three transformation experiments performed in four different strains. Growth curves were repeated at least three times with three replicates each. All attempts at replication were successful. |
| Randomization | Not applicable since samples were not allocated to experimental groups. |
| Blinding | Not applicable since samples were not allocated to experimental groups. |

# Reporting for specific materials, systems and methods

We require information from authors about some types of materials, experimental systems and methods used in many studies. Here, indicate whether each material, system or method listed is relevant to your study. If you are not sure if a list item applies to your research, read the appropriate section before selecting a response.

## Materials & experimental systems

| n/a | Involved in the study |
|---|---|
| ☒ | ☐ Antibodies |
| ☒ | ☐ Eukaryotic cell lines |
| ☒ | ☐ Palaeontology and archaeology |
| ☒ | ☐ Animals and other organisms |
| ☒ | ☐ Human research participants |
| ☒ | ☐ Clinical data |
| ☒ | ☐ Dual use research of concern |

## Methods

| n/a | Involved in the study |
|---|---|
| ☒ | ☐ ChIP-seq |
| ☒ | ☐ Flow cytometry |
| ☒ | ☐ MRI-based neuroimaging |

