## [Peer Review File · Nature Microbiology]

Peer Review Information

Journal: Nature Microbiology

Manuscript Title: A bacterial pan-genome makes gene essentiality strain-dependent and evolvable

Corresponding author name(s): Tim van Opijnen

Reviewer Comments & Decisions:

Decision Letter, initial version:

Dear Professor van Opijnen,

Thank you for your patience while your manuscript "A bacterial pan-genome makes gene essentiality strain-dependent and evolvable" was under peer-review at Nature Microbiology. It has now been seen by 3 referees, whose expertise and comments you will find at the of this email. You will see from their comments below that while they find your work of interest, some important points are raised. We are very interested in the possibility of publishing your study in Nature Microbiology, but would like to consider your response to these concerns in the form of a revised manuscript before we make a final decision on publication.

In particular, you will see that referees are positive about your work. Referee #2 does not raise any issue, and referee #3 has only minor points that should be easily addressed. Referee #1 raise a few specific points, including the suggestion to complement your isolates selection with more isolates representing additional clades. The rest referees' reports are clear and the remaining issues should be straightforward to address.

If you have not done so already please begin to revise your manuscript so that it conforms to our Article format instructions at <http://www.nature.com/nmicrobiol/info/final-submission/>

The usual length limit for a Nature Microbiology Article is six display items (figures or tables) and 3,000 words. We have some flexibility, and can allow a revised manuscript at 3,500 words, but please consider this a firm upper limit. There is a trade-off of ~250 words per display item, so if you need more space, you could move a Figure or Table to Supplementary Information.

Some reduction could be achieved by focusing any introductory material and moving it to the start of your opening 'bold' paragraph, whose function is to outline the background to your work, describe in a sentence your new observations, and explain your main conclusions. The discussion should also be limited. Methods should be described in a separate section following the discussion, we do not place a word limit on Methods.

Nature Microbiology titles should give a sense of the main new findings of a manuscript, and should not contain punctuation. Please keep in mind that we strongly discourage active verbs in titles, and that they should ideally fit within 90 characters each (including spaces).

2We strongly support public availability of data. Please place the data used in your paper into a public data repository, if one exists, or alternatively, present the data as Source Data or Supplementary Information. If data can only be shared on request, please explain why in your Data Availability Statement, and also in the correspondence with your editor. For some data types, deposition in a public repository is mandatory - more information on our data deposition policies and available repositories can be found at <https://www.nature.com/nature-research/editorial-policies/reporting-standards#availability-of-data>.

Please include a data availability statement as a separate section after Methods but before references, under the heading "Data Availability". This section should inform readers about the availability of the data used to support the conclusions of your study. This information includes accession codes to public repositories (data banks for protein, DNA or RNA sequences, microarray, proteomics data etc...), references to source data published alongside the paper, unique identifiers such as URLs to data repository entries, or data set DOIs, and any other statement about data availability. At a minimum, you should include the following statement: "The data that support the findings of this study are available from the corresponding author upon request", mentioning any restrictions on availability. If DOIs are provided, we also strongly encourage including these in the Reference list (authors, title, publisher (repository name), identifier, year). For more guidance on how to write this section please see:

<http://www.nature.com/authors/policies/data/data-availability-statements-data-citations.pdf>

To improve the accessibility of your paper to readers from other research areas, please pay particular attention to the wording of the paper's opening bold paragraph, which serves both as an introduction and as a brief, non-technical summary in about 150 words. If, however, you require one or two extra sentences to explain your work clearly, please include them even if the paragraph is over-length as a result. The opening paragraph should not contain references. Because scientists from other sub-disciplines will be interested in your results and their implications, it is important to explain essential but specialised terms concisely. We suggest you show your summary paragraph to colleagues in other fields to uncover any problematic concepts.

If your paper is accepted for publication, we will edit your display items electronically so they conform to our house style and will reproduce clearly in print. If necessary, we will re-size figures to fit single or double column width. If your figures contain several parts, the parts should form a neat rectangle when assembled. Choosing the right electronic format at this stage will speed up the processing of your paper and give the best possible results in print. We would like the figures to be supplied as vector files - EPS, PDF, AI or postscript (PS) file formats (not raster or bitmap files), preferably generated with vector-graphics software (Adobe Illustrator for example). Please try to ensure that all figures are non-flattened and fully editable. All images should be at least 300 dpi resolution (when figures are scaled to approximately the size that they are to be printed at) and in RGB colour format. Please do not submit Jpeg or flattened TIFF files. Please see also 'Guidelines for Electronic Submission of Figures' at the end of this letter for further detail.

Figure legends must provide a brief description of the figure and the symbols used, within 350 words, including definitions of any error bars employed in the figures.

2When submitting the revised version of your manuscript, please pay close attention to our [Digital Image Integrity Guidelines](https://www.nature.com/nature-research/editorial-policies/image-integrity) and to the following points below:

Please include a statement before the acknowledgements naming the author to whom correspondence and requests for materials should be addressed.

Finally, we require authors to include a statement of their individual contributions to the paper -- such as experimental work, project planning, data analysis, etc. -- immediately after the acknowledgements. The statement should be short, and refer to authors by their initials. For details please see the Authorship section of our joint Editorial policies at http://www.nature.com/authors/editorial_policies/authorship.html

- * include a point-by-point response to any editorial suggestions and to our referees. Please include your response to the editorial suggestions in your cover letter, and please upload your response to the referees as a separate document.
- * ensure it complies with our format requirements for Letters as set out in our guide to authors at www.nature.com/nmicrobiol/info/gta/
- * state in a cover note the length of the text, methods and legends; the number of references; number and estimated final size of figures and tables
- * resubmit electronically if possible using the link below to access your home page:

{redacted}

*This url links to your confidential homepage and associated information about manuscripts you may have submitted or be reviewing for us. If you wish to forward this e-mail to co-authors, please delete this link to your homepage first.

Please ensure that all correspondence is marked with your Nature Microbiology reference number in the subject line.

Nature Microbiology is committed to improving transparency in authorship. As part of our efforts in this direction, we are now requesting that all authors identified as 'corresponding author' on published papers create and link their Open Researcher and Contributor Identifier (ORCID) with their account on the Manuscript Tracking System (MTS), prior to acceptance. This applies to primary research papers only. ORCID helps the scientific community achieve unambiguous attribution of all scholarly contributions. You can create and link your ORCID from the home page of the MTS by clicking on 'Modify my Springer Nature account'. For more information please visit www.springernature.com/orcid.

We hope to receive your revised paper within three weeks. If you cannot send it within this time, please let us know.

Yours sincerely,

{redacted}

Reviewer Expertise:

Referee #1: S. pneumoniae/pangenome analyses/Tn-seq/RNA-seq

Referee #2: pangenome

Referee #3: S. pneumoniae

Reviewers Comments:

Reviewer #1 (Remarks to the Author):

This manuscript covers an extensive analysis of essential genes within the pneumococcal pangenome, using 36 clinical isolates representative of ~2/3 (see caveats below) of the pangenome, and integrating whole genome sequencing (36 genomes resequenced using long reads), RNA-seq (36 isolates), and Tn-seq (21 isolates) technologies. The authors identified 206 universally essential genes (i.e. essential in all isolates studied, the commonly accepted terminology), but also 186 strain-specific essential genes (always present but not always essential), and 128 accessory essential genes (essential when present). They further demonstrate that the genetic background influences gene essentiality, but may not entirely compensate essentiality, i.e. genes still impose a fitness cost.

Initial in silico identifications of the three types of essential genes were experimentally assayed using

4a panoply of approaches, including fitness in vitro and in vivo, genetic background-dependent evolvability of essential genes in vitro, and a substantial panel of functional, multi-gene interaction, and/or drug testing assays for different essential genes (Figures 4, 5, 6, 7, and 8). These experiments reveal new aspects of pneumococcal biology, but also that some drugs' effects may be undermined in certain genetic backgrounds among a bacterial population's pangenome.

Overall, this systems biology study is very comprehensive and rich in complex figures, yet easy to follow. A few specific questions are listed below.

Line 83 and elsewhere: the designation "strain-specific essential genes" is misleading since these are genes present in all isolates. Integrating the term "core gene" or a notion that these genes are present in all isolates in this description and that of universally essential genes would provide for a more unequivocal framework.

Lines 106-117: the selection of the 111 worldwide isolates and the 38 isolates predating 1974 is interesting, especially from a geographical perspective, but does not guarantee that the entire genomic diversity of the pneumococcal population will be represented. It would be best to complement these selections with isolates representing potential additional clades derived from a whole genome-based phylogeny from an unbiased, much larger selection of pneumococcal genomes among the thousands of genomes currently available. In addition, some of the clades on the limited tree presented in Figure 1 are not well represented by the 36 isolates selected for this study (that are biased towards the collection from the Netherlands likely directly available to the authors). The text does indicate >68% coverage of the pangenome, thus not full coverage, but this is likely an underestimation based on the first comment in this paragraph.

Lines 159-162: How many unique genes total, of the 520, or 206+186, are represented in these 5 categories? This number can't be inferred from Figure 2F since a given gene might belong to multiple categories.

Lines 502-505: the terms "untangle" and "unpacked" are not specific or explicit. The sentences should be reworded to clarify the meaning, including that strain-dependent essential genes offer a unique opportunity to study their function in genetic backgrounds where they are not essential, and the opportunity to identify and characterize compensatory mutations.

Reviewer #2 (Remarks to the Author):

I think this is a remarkable paper, full of insight and detail. The work is well-designed. The authors take a representative sample of the known pangenome of *Streptococcus pneumoniae* genomes from around the world and performed a variety of analyses. Using Tn-Seq in order to explore gene essentiality, they provide insights into strain-specific essential genes - the kinds of genes that might be dismissed as being non-essential, before this paper. The authors not only record the results of these experiments, but also go further into trying to understand how seemingly essential genes, when disrupted, become dispensable. They unearth mechanisms, SNPs and contexts in which essentiality becomes fluid.

5I am in the rare situation where I don't have any comments. There might be one or two places where the writing might be tightened up, but overall, this is a very carefully executed study that benefits from excellent study design and a comprehensive set of follow-up experiments.

I feel that this paper will be transformational in the study of pangenomes and in our understanding of what is essential and what is dispensable in a genome, and when genes might change from being one to being the other.

- James McInerney (reviewer).

Reviewer #3 (Remarks to the Author):

The manuscript entitled "A bacterial pan-genome makes gene essentiality strain-dependent and evolvable" by Rosconi et al not only addresses a theme of high relevance, but it does so by using an elegant blend of traditional and high-throughput approaches using the pneumococcal pangenome as a reference.

The pangenome was defined out of more than 200 pneumococcal isolates, from which a set of 36 strains representing almost 70% of the pangenome were used for experimental analysis. Transposon libraries enabling high-confidence essentiality predictions were obtained for 17 of the 36 strains. This enabled the identification of three categories of essential genes: universal, strain-dependent and accessory essential genes. Further, the authors found using a forced-evolution experiment that strain-dependent essential genes could evolve into non-essential genes. Moreover, examples for how accessory genes can influence the essentialome were identified.

Essential gene products represent attractive targets for developing antimicrobials, making this study highly relevant. Although mapping essential genes has been the focus of numerous studies before, the authors in this study go beyond what is already known about essentiality in *S. pneumoniae*. By exploring the issue in the lens of a large pangenome, the study revealed the evolvability of essential genes and mechanisms influencing essentiality.

Minor comments:

1. Conditional essentiality (eg. growth in vivo vs in vitro) could be discussed in relation to the PG-collection, since screening was only done under laboratory conditions in defined medium.
2. L.102. It is not clear what the authors mean by "genetically similar pairs".
3. L. 125. TA in the context of dinucleotides could be explained, since the authors use the abbreviation TA for teichoic acids later in the text.
4. It would be interesting to read what the authors think about the potential biases in using Tn-Seq as an approach to find essential genes, since this can only be used for strains that can be transformed under laboratory conditions (Eg. from the PG36, only 17 had optimal saturation).
 - L.200. Have the authors tested whether the merodiploid colonies could be resolved by allowing them to further grow in fresh medium with the selective antibiotic.
 - L. 558. Could the authors present a reference for the defined media used in the experiments?
 - L. 676. The full form for TPM abbreviation is missing.
 - L. 786. Not clear in the legend if TIGR4 was used for both conditions.
 - L.829 and 835. Have the authors used a post-hoc test with the Ordinary One-Way ANOVA?

6Reviewer #1 (Remarks to the Author)

Reviewer 1 - Comment 1.

“Line 83 and elsewhere: the designation “strain-specific essential genes” is misleading since these are genes present in all isolates. Integrating the term “core gene” or a notion that these genes are present in all isolates in this description and that of universally essential genes would provide for a more unequivocal framework.”

Author response:

We have made the categories and distinctions between them clearer by defining them as: Universal essentials (core genome genes that are present and essential in each strain), Core strain-dependent essentials (core genome genes that are present in all strains but only essential in some), and Accessory essentials (accessory genome genes essential in strains where they are present). This new designation should resolve ambiguity associated with strain-dependent essentials, and we thank the reviewer for pointing this out.

Reviewer 1 - Comment 2.

“Lines 106-117: the selection of the 111 worldwide isolates and the 38 isolates predating 1974 is interesting, especially from a geographical perspective, but does not guarantee that the entire genomic diversity of the pneumococcal population will be represented. It would be best to complement these selections with isolates representing potential additional clades derived from a whole genome-based phylogeny from an unbiased, much larger selection of pneumococcal genomes among the thousands of genomes currently available. In addition, some of the clades on the limited tree presented in Figure 1 are not well represented by the 36 isolates selected for this study (that are biased towards the collection from the Netherlands likely directly available to the authors). The text does indicate >68% coverage of the pangenome, thus not full coverage, but this is likely an underestimation based on the first comment in this paragraph.”

Author response:

We understand the reviewer's comment, which we believe is partially caused by some confusion in the manner strains were selected and their pan-genome coverage was calculated. Initially we carefully selected 15 strain pairs that scatter evenly through a phylogenetic tree that was constructed from a surveillance genome-sequencing study consisting of >350 invasive strains isolated before and after PCV7 vaccine implementation¹. Considering that all the strains were isolated from the same location, we aimed to determine whether this phylogenetic diversity was robust when compared against worldwide isolates/diversity. To accomplish this, we carefully selected a set of 178 isolates from multiple surveillances studies around the world, resulting in our pan-genome study group. Note that in the 2 paragraphs below we highlight in detail how this group was put together. Importantly, Extended Data Figure 1a shows that the 15-strain pairs distribute in all the major branches of the world-wide genomes constructed tree, which shows that our collection is not restricted to a putative "Nijmegen-specific" clade, and thus the collection reflects world-wide diversity. Moreover, the reviewer's comment inspired us to test and include how our dataset fits with the clusters from the Global Pneumococcal Sequencing Project database. With this new analysis we show that our collection represents 70 Global Pneumococcal Sequencing clusters. Moreover, the GPS clusters covered by PG-collection, contains isolates from different parts of the world.

Selecting 208 isolates:

The 208 isolates were selected as follows: to the 30 strains selected from the Nijmegen surveillance, we added six diverse clinical and experimental lab strains we had available at the lab, TIGR4 (first *S. pneumoniae* sequenced strain isolated from Norway), Taiwan-19F (a penicillin-resistant strain isolated from Taiwan), BHN97 (isolated from Norway), two recent US isolates (CT22F and GA22F), and a derived clone of the Griffith-Avery model strain D39. To include diversity obtained from various large-scale survey /sequencing projects that were performed across the world we added 30 strains from a set of isolates collected from children during routine visits to primary care physicians in Massachusetts, US², 30 from a refugee camp in Maela, Thailand³, 30 from Southampton (UK) ("Whole genome sequencing of carried *Streptococcus pneumoniae* during the implementation of pneumococcal conjugate vaccines in the UK", Bioproject PRJEB2417), and 21 from the Malawi-Liverpool-Wellcome Trust Clinical Research Programme (MLW) strain archive⁴. Furthermore, we added the 23 *S. pneumoniae* closed genomes available in RefSeq⁵, and completed the set with the 38 strains isolated before 1974⁶. Note that from each project similar numbers of strains were chosen to add to our 'Nijmegen-collection' in order to obtain a well-balanced dataset, that is not dominated by a single study. This set of 208 isolates should thus be a good representative of the currently described *S. pneumoniae* diversity. Moreover, the compact size enables easy visualization,

manipulation and calculations of for instance the pan-genome, what type of genes are included and what our dataset covers.

To determine what genes our dataset includes we divided the pan-genome in core and accessory genome. Moreover, the accessory genome could be further subdivided in “shell” (genes present in most isolates but not all) and “cloud” genes (genes present only in a very small number of isolates). Our Supplementary tables 3 and 7 highlight that from the 1205 (32%) pan-genome clusters not represented in our PG-collection, 55% of them are only present in less than 5 out of the 208 genomes (Extended Data Figure 1b). This shows that our collection misses mostly “cloud” genes. And while some of these may be important for some strains, and likely under very specific conditions, we expect that, due to their low frequency, their importance for the pan-genome is limited.

Finally, to further confirm that our 208-strain study-group is a good representative of the *S. pneumoniae* pan-genome, we performed, and added to the manuscript, a rarefaction analysis (Extended Data Figure 1c). This analysis shows that our pan-genome reaches a plateau, indicating that adding more genomes to our study-group would mostly add more “cloud” genes. However, while we think that research focused on “cloud” genes, or for that matter clades not represented in our collection, could add important knowledge on the pan-genome as well as *S. pneumoniae* biology, this would be beyond the scope of this work. Importantly, our aim was to combine comparative genomics with ‘wet-bench’ experiments, using a feasible number of strains that represent *S. pneumoniae* biology at the widest possible level. While we believe that we successfully reached that goal, we think that thanks to the reviewer’s comment, the description of how strains were selected and what they cover pan-genome wide has been adjusted in the manuscript and is much clearer. Most importantly, it further confirms we cover the majority of the diversity that is present within *S. pneumoniae*’s pan-genome, while if we only consider the shell accessory genome, our ~68% coverage may even be an underestimation.

To highlight and clarify our original and additional analyses we have updated the manuscript in multiple places. These changes/additions are as follows:

- 1) Global Pneumococcal Sequencing Project database clusters covered by the PG-collection and the 208 strains pan-genome study group are now included in supplementary tables 1 and the new supplementary table 3. We also describe the GPS clusters coverage in the main text, page 5, lines 78-96.
- 2) Extended Data Figure 1b shows that genes not represented in the PG-collection are mostly “cloud” genes.
- 3) Extended Data Figure 1c shows the rarefaction analysis of the 208 selected strains, which indicates that pan-genome gene numbers reach a plateau after ~80 isolates. Moreover, this plateau is similar as previously described^{7,8}.

Reviewer 1 - Comment 3.

“Lines 159-162: How many unique genes total, of the 520, or 206+186, are represented in these 5 categories? This number can’t be inferred from Figure 2F since a given gene might belong to multiple categories.”

Author response:

We apologize for the figure not showing the data. To solve this, we added a new table (supplementary table 8) which contains the number of genes for each essentialome class belonging to the different categories, including the p-values and adjusted p-values obtained from the enrichment analysis.

Reviewer 1 - Comment 4.

“Lines 502-505: the terms “untangle” and “unpacked” are not specific or explicit. The sentences should be reworded to clarify the meaning, including that strain-dependent essential genes offer a unique opportunity to study their function in genetic backgrounds where they are not essential, and the opportunity to identify and characterize compensatory mutations.”

Author response:

In order to fit the journal’s format, we have rewritten multiple sections and made sure neither “untangle” nor “unpacked” remains in the new version. The final sentence that explains the same idea in the discussion (page 13, lines 328-330) now states: *“An advantage of this strain-dependent essentiality is that it enables an approach to infer gene function for instance through analyzing compensatory mechanisms.”*

Reviewer #2 (Remarks to the Author)

Reviewer 2 had no comments.

Reviewer #3 (Remarks to the Author):

Reviewer 3.

No Major Comments

Minor comments

Reviewer 3. – Minor Comment 1:

“Conditional essentiality (eg. growth in vivo vs in vitro) could be discussed in relation to the PG-collection, since screening was only done under laboratory conditions in defined medium.”

Author response:

A gene is categorized as conditionally essential when it is required for growth in one or more specific conditions. For instance, tryptophan biosynthesis genes are essential only if there is no tryptophan source available in the environment. Transposon mutant libraries are generated in rich medium to ensure environmental stress is minimal and genes identified as essential, i.e., those genes absolutely required for growth under any condition are maximized. It remains possible we identify genes as essential, while in reality they are conditionally-essential but only under highly specialized environmental conditions, however this remains a point of discussion for every essential gene ever identified. Moreover, it will be extremely difficult, time-consuming and possibly impossible to determine the exact condition that any essential-gene may potentially be conditionally-essential. Importantly, in this work the main goal is to determine the importance and influence of the genetic-background on gene essentiality. We base our strain-specific essentiality on our *in vitro* Tn-Seq data; this results in some genes being called essential in all strains while others are essential only in certain strains. It is possible that some

11genes may be essential *in vitro*, while non-essential *in vivo*. Such a gene should of course be called conditionally-essential, and we would not identify them with our approach. However, while possible, and very interesting, such a gene will likely be incredibly rare. Unfortunately, we have had to significantly shorten the discussion of our manuscript and there is no room to discuss these kinds of essentiality-instances in detail. However, we have made sure to clarify throughout the manuscript the difference between essentiality and conditional essentiality, the importance of conditionally essential and strain-specific essential genes *in vivo*, and also highlight how strain-specific essential genes can be used to untangle gene-function and thereby possibly identify more conditionally-essential genes.

Reviewer 3. – Minor Comment 2:

“L.102. It is not clear what the authors mean by “genetically similar pairs”.”

Author response:

Genetically similar pairs refer to two strains that are highly similar at the genomic level. This means: 1) a phylogeny based on SNPs in core genes categorizes the isolates as phylogenetic pairs (Extended Data Figure 1a); and 2) the accessory genomic content of both isolates is almost identical. To further clarify this, we added a new table (supplementary table 2) with the Average Nucleotide Identity (ANI) for all strain-to-strain comparisons. These genetically similar pairs have ANI values > 99.1. The methodology to determine ANI values is now described in the methods section (P15L399-400).

Reviewer 3. – Minor Comment 3:

“L. 125. TA in the context of dinucleotides could be explained, since the authors use the abbreviation TA for teichoic acids later in the text.”

Author response:

We have made changes to the main text by using “teichoic acids” instead of “TA”. Only Figure 3 and Extended Data Figure 5 still use TA but we made this clear in the figure legend where we also use and define LTA and WTA.

Reviewer 3. – Minor Comment 4:

“It would be interesting to read what the authors think about the potential biases in using Tn-Seq as an approach to find essential genes, since this can only be used for strains that can be transformed under laboratory conditions (Eg. from the PG36, only 17 had optimal saturation).”

Author response:

Using Tn-Seq to identify essential genes is based on the likelihood that the absence of Tn-insertions in a specific gene is due to a gene’s essentiality, taking into account the saturation of the library, the size of a gene and number of TA-sites (target-sites) in a gene. Missing insertions could thus be due to a gene’s essentiality; however, it could also be due to chance, which could be partially affected by a strain’s transformability (transformation efficiency), or even the accessibility/susceptibility to a Tn-insertion of a specific genomic location. We have previously shown that there is very little bias in where the transposon inserts itself in the genome^{9,10}. However, transformation efficiency can be widely different between strains. For instance, restriction-modification systems and capsule-types have been described as important for affecting transformation efficiency. However, many questions as to what can additionally drive efficiency remain unanswered. In our collection even highly-similar pairs can have very different efficiencies, e.g., we obtained the highest saturated libraries for PG04, but no transformants for the highly similar strain PG03. This suggests that small (unknown) differences can significantly affect an isolate’s transformability. While Tn-Seq may miss or miss-classify some essential genes, our approach of using so many different strains, strain-pairs and combined with our many validation experiments (including expression levels, fitness effects, sequence diversity and gene knockout and forced evolution validations) indicate those miss-classifications will be rare. An alternative approach to Tn-Seq would be using CRISPRi knockdown libraries. While this could possibly further improve the certainty of each essentiality call for each gene in each strain, constructing and validating CRISPRi guide-RNA pools for multiple strains is expensive and time consuming. Moreover, also for CRISPRi one would need efficiently transformable strains. Due to manuscript length limitations we are not able to discuss this in more detail. However, the way Tn-Seq is used to make essential-gene calls is explained in detail.

Reviewer 3. – Minor Comment 5:

“L.200. Have the authors tested whether the merodiploid colonies could be resolved by allowing them to further grow in fresh medium with the selective antibiotic.”

Author response:

This is a great experiment and something we would love to do. It would be interesting to see whether merodiploids could be ‘unstable’ and lead to new function and/or essential gene bypass solutions. However, what would be key for a new solution/function to emerge from a merodiploid (at least in a reasonable experimental timeline) is that the merodiploid would create some kind of selectable defect, i.e., one that would create a selection pressure that can be overcome over time by further innovation. In our experiments, we found that merodiploid growth did not register a defect. Then, although evolutionarily possible, it would likely take too many passages and time to obtain a resolved merodiploid that could outcompete the growth of the entire merodiploid population. We agree it is a very interesting idea and possibly doable in alternative ways, but is beyond the scope of the current manuscript.

Reviewer 3. – Minor Comment 5:

“L. 558. Could the authors present a reference for the defined media used in the experiments?”

Author response:

We added the composition of SDMM as supplementary data.

Reviewer 3. – Minor Comment 6:

“L. 676. The full form for TPM abbreviation is missing.”

Author response:

‘Transcript per millions’ is added in the Methods section (page 17, line 476).

Reviewer 3. – Minor Comment 7:

“L. 786. Not clear in the legend if TIGR4 was used for both conditions.”

Author response:

We clarified in the legend that SDMM Tn-Seq results include all strains with libraries.

Reviewer 3. – Minor Comment 8:

“L.829 and 835. Have the authors used a post-hoc test with the Ordinary One-Way ANOVA?”

Author response:

In the case of lipid composition experiments, an ANOVA was performed for each lipid, while comparing the levels of each WT strain to their derived knockout. In the case of PtdGro we also compared between WT strains. Since the number of comparisons is low, we decided not to correct for multiple comparisons but applied an uncorrected Fisher’s LSD test, which we have now clarified in the manuscript. In the case of growth rate and maximum OD, we compared wild types against derived knockouts also using the uncorrected Fisher’s LSD test, which has also been clarified

Decision Letter, first revision:

Dear Dr. van Opijnen,

Thank you for submitting your revised manuscript "A bacterial pan-genome makes gene essentiality strain-dependent and evolvable" (NMICROBIOL-22020377A). It has now been seen by the original referees and their comments are below. The reviewers find that the paper has improved in revision, and therefore we'll be happy in principle to publish it in Nature Microbiology, pending minor revisions to satisfy the referees' final requests and to comply with our editorial and formatting guidelines.

Thank you again for your interest in Nature Microbiology Please do not hesitate to contact me if you have any questions.

Sincerely,

15--
{redacted}

Reviewer #1 (Remarks to the Author):

The authors have provided very detailed and insightful responses to each of the reviewers' comments, and given manuscript length limitations, edited the main text to some extent but included additional supplemental data, tables, and/or figures that definitely help understand and justify the research presented, including the important aspect of genome and strain selection for pangenome and experimental analyses, respectively. In addition, the new naming scheme for the three categories of essential genes is now to the point and unequivocal.

The glossary of technical terms used throughout the text is also helpful. It would probably be worth including the detailed description of the choice of strains and genomes for the study that was provided in the rebuttal letter as supplemental text, along with any other relevant specifics (like the glossary).

The already excellent and comprehensive study initially presented has been further improved to avoid confusion and remedy the partial lack of supporting details/data, resulting in an impactful report challenging the classical concept of gene essentiality.

Reviewer #3 (Remarks to the Author):

The authors have addressed all comments in this well-written and beautifully designed study.

Decision Letter, final checks:

Dear Dr. van Opijnen,

Thank you for your patience as we've prepared the guidelines for final submission of your Nature Microbiology manuscript, "A bacterial pan-genome makes gene essentiality strain-dependent and evolvable" (NMICROBIOL-22020377A). Please carefully follow the step-by-step instructions provided in the attached file, and add a response in each row of the table to indicate the changes that you have made. Please also check and comment on any additional marked-up edits we have proposed within the text. Ensuring that each point is addressed will help to ensure that your revised manuscript can be swiftly handed over to our production team.

16When you upload your final materials, please include a point-by-point response to any remaining reviewer comments.

In recognition of the time and expertise our reviewers provide to Nature Microbiology's editorial process, we would like to formally acknowledge their contribution to the external peer review of your manuscript entitled "A bacterial pan-genome makes gene essentiality strain-dependent and evolvable". For those reviewers who give their assent, we will be publishing their names alongside the published article.

Nature Microbiology offers a Transparent Peer Review option for new original research manuscripts submitted after December 1st, 2019. As part of this initiative, we encourage our authors to support increased transparency into the peer review process by agreeing to have the reviewer comments, author rebuttal letters, and editorial decision letters published as a Supplementary item. When you submit your final files please clearly state in your cover letter whether or not you would like to participate in this initiative. Please note that failure to state your preference will result in delays in accepting your manuscript for publication.

Cover suggestions

As you prepare your final files we encourage you to consider whether you have any images or illustrations that may be appropriate for use on the cover of Nature Microbiology.

Nature Microbiology has now transitioned to a unified Rights Collection system which will allow our Author Services team to quickly and easily collect the rights and permissions required to publish your work. Approximately 10 days after your paper is formally accepted, you will receive an email in

17providing you with a link to complete the grant of rights. If your paper is eligible for Open Access, our Author Services team will also be in touch regarding any additional information that may be required to arrange payment for your article.

Please note that *Nature Microbiology* is a Transformative Journal (TJ). Authors may publish their research with us through the traditional subscription access route or make their paper immediately open access through payment of an article-processing charge (APC). Authors will not be required to make a final decision about access to their article until it has been accepted. [Find out more about Transformative Journals](https://www.springernature.com/gp/open-research/transformative-journals)

Authors may need to take specific actions to achieve [compliance with funder and institutional open access mandates](https://www.springernature.com/gp/open-research/funding/policy-compliance-faqs). If your research is supported by a funder that requires immediate open access (e.g. according to [Plan S principles](https://www.springernature.com/gp/open-research/plan-s-compliance)) then you should select the gold OA route, and we will direct you to the compliant route where possible. For authors selecting the subscription publication route, the journal's standard licensing terms will need to be accepted, including [self-archiving policies](https://www.nature.com/nature-portfolio/editorial-policies/self-archiving-and-license-to-publish). Those licensing terms will supersede any other terms that the author or any third party may assert apply to any version of the manuscript.

Please use the following link for uploading these materials:
{redacted}

Best regards,

{redacted}

--

Reviewer #1:

18Remarks to the Author:

The authors have provided very detailed and insightful responses to each of the reviewers' comments, and given manuscript length limitations, edited the main text to some extent but included additional supplemental data, tables, and/or figures that definitely help understand and justify the research presented, including the important aspect of genome and strain selection for pangenome and experimental analyses, respectively. In addition, the new naming scheme for the three categories of essential genes is now to the point and unequivocal.

The glossary of technical terms used throughout the text is also helpful. It would probably be worth including the detailed description of the choice of strains and genomes for the study that was provided in the rebuttal letter as supplemental text, along with any other relevant specifics (like the glossary).

The already excellent and comprehensive study initially presented has been further improved to avoid confusion and remedy the partial lack of supporting details/data, resulting in an impactful report challenging the classical concept of gene essentiality.

Reviewer #3:

Remarks to the Author:

The authors have addressed all comments in this well-written and beautifully designed study.

Final Decision Letter:

Dear Professor van Opijnen,

I am pleased to accept your Article "A bacterial pan-genome makes gene essentiality strain-dependent and evolvable" for publication in Nature Microbiology. Thank you for having chosen to submit your work to us and many congratulations.

19Due to the importance of these deadlines, we ask you please us know now whether you will be difficult to contact over the next month. If this is the case, we ask you provide us with the contact information (email, phone and fax) of someone who will be able to check the proofs on your behalf, and who will be available to address any last-minute problems.

Acceptance of your manuscript is conditional on all authors' agreement with our publication policies (see <https://www.nature.com/nmicrobiol/editorial-policies>). In particular your manuscript must not be published elsewhere and there must be no announcement of the work to any media outlet until the publication date (the day on which it is uploaded onto our website).

Please note that *Nature Microbiology* is a Transformative Journal (TJ). Authors may publish their research with us through the traditional subscription access route or make their paper immediately open access through payment of an article-processing charge (APC). Authors will not be required to make a final decision about access to their article until it has been accepted. [Find out more about Transformative Journals](https://www.springernature.com/gp/open-research/transformative-journals)

Authors may need to take specific actions to achieve [compliance with funder and institutional open access mandates](https://www.springernature.com/gp/open-research/funding/policy-compliance-faqs). If your research is supported by a funder that requires immediate open access (e.g. according to [Plan S principles](https://www.springernature.com/gp/open-research/plan-s-compliance)) then you should select the gold OA route, and we will direct you to the compliant route where possible. For authors selecting the subscription publication route, the journal's standard licensing terms will need to be accepted, including [self-archiving policies](https://www.nature.com/nature-portfolio/editorial-policies/self-archiving-and-license-to-publish). Those licensing terms will supersede any other terms that the author or any third party may assert apply to any version of the manuscript.

We welcome the submission of potential cover material (including a short caption of around 40 words) related to your manuscript; suggestions should be sent to Nature Microbiology as electronic files (the image should be 300 dpi at 210 x 297 mm in either TIFF or JPEG format). Please note that such pictures should be selected more for their aesthetic appeal than for their scientific content, and that colour images work better than black and white or grayscale images. Please do not try to design a cover with the Nature Microbiology logo etc., and please do not submit composites of images related

20to your work. I am sure you will understand that we cannot make any promise as to whether any of your suggestions might be selected for the cover of the journal.
